# Brixia and qSOFA Scores, Coagulation Factors and Blood Values in Spring versus Autumn 2021 Infection in Pregnant Critical COVID-19 Patients: A Preliminary Study

**DOI:** 10.3390/healthcare10081423

**Published:** 2022-07-29

**Authors:** Catalina Filip, Roxana Covali, Demetra Socolov, Mona Akad, Alexandru Carauleanu, Ingrid Andrada Vasilache, Ioana Sadiye Scripcariu, Ioana Pavaleanu, Tudor Butureanu, Madalina Ciuhodaru, Lucian Vasile Boiculese, Razvan Socolov

**Affiliations:** 1Department of Vascular Surgery, Grigore T. Popa University of Medicine and Pharmacy Iasi, 700115 Iasi, Romania; filipcatalina20@gmail.com; 2Department of Radiology, Elena Doamna Obstetrics and Gynecology University Hospital, Grigore T. Popa University of Medicine and Pharmacy Iasi, 700115 Iasi, Romania; 3Department of Obstetrics and Gynecology, Cuza Voda Obstetrics and Gynecology University Hospital, Grigore T. Popa University of Medicine and Pharmacy Iasi, 700115 Iasi, Romania; demetra.socolov@umfiasi.ro (D.S.); acarauleanu@yahoo.com (A.C.); ingrid-andrada-n-tanasa@d.umfiasi.ro (I.A.V.); isscripcariu@gmail.com (I.S.S.); 4Department of Obstetrics and Gynecology, Grigore T. Popa University of Medicine and Pharmacy Iasi, 700115 Iasi, Romania; 5Department of Obstetrics and Gynecology, Elena Doamna Obstetrics and Gynecology University Hospital, Grigore T. Popa University of Medicine and Pharmacy Iasi, 700115 Iasi, Romania; ioana-m-pavaleanu@umfiasi.ro (I.P.); tudorandreib@gmail.com (T.B.); mciuhodaru@yahoo.com (M.C.); socolovr@yahoo.com (R.S.); 6Department of Statistics, Grigore T. Popa University of Medicine and Pharmacy Iasi, 700115 Iasi, Romania; lboiculese@gmail.com

**Keywords:** SARS-CoV-2, variants of concern, alpha variant, delta variant, Brixia score, qSOFA score, GGT, LDH, CRP, pregnancy

## Abstract

(1) Background: From the recent variants of concern of the SARS-CoV-2 virus, in which the delta variant generated more negative outcomes than the alpha, we hypothesized that lung involvement, clinical condition deterioration and blood alterations were also more severe in autumn infection, when the delta variant dominated (compared with spring infections, when the alpha variant dominated), in severely infected pregnant patients. (2) Methods: In a prospective study, all pregnant patients admitted to the ICU of the Elena Doamna Obstetrics and Gynecology Hospital with a critical form of COVID-19 infection—spring group (n = 11) and autumn group (n = 7)—between 1 January 2021 and 1 December 2021 were included. Brixia scores were calculated for every patient: A score, upon admittance; H score, the highest score throughout hospitalization; and E score, at the end of hospitalization. For each day of Brixia A, H or E score, the qSOFA (quick sepsis-related organ failure assessment) score was calculated, and the blood values were also considered. (3) Results: Brixia E score, C-reactive protein, GGT and LDH were much higher, while neutrophil count was much lower in autumn compared with spring critical-form pregnant patients. (4) Conclusions: the autumn infection generated more dramatic alterations than the spring infection in pregnant patients with critical forms of COVID-19. Larger studies with more numerous participants are required to confirm these results.

## 1. Introduction

Of the recent variants of concern of the SARS-CoV-2 virus, two of them were devastating all around the world: the alpha variant, with a much higher transmissibility than previous ones [1], and the delta variant, which affected even fully vaccinated persons [2]. The delta variant generated more negative outcomes than the alpha in the general population and in the pregnant population. We hypothesize that lung involvement, clinical condition deterioration and blood alterations are also more severe in autumn infection, when the delta variant dominated, compared with spring infections, when the alpha variant dominated, in SARS-CoV-2 critical-infected pregnant patients.

Lung involvement can be evaluated primarily by CT or, in facilities without CT or in ICUs (intensive care units), by conventional chest X-rays. The use of mobile radiology installations allows pulmonary examination in patients in the ICU with minimal position change and without needing to disconnect the patient from life-supporting devices.

Mild and moderate cases of COVID-19 may show no abnormality on chest X-rays, while in severe cases, opacities are visible in both lungs—either interstitial and/or alveolar-type opacities. These opacities are situated to the periphery of the lungs, in lower lobes, and have a tendency to unite. With proper therapy, opacities slowly disappear, but sometimes, they involve the majority of the lung area with a catastrophic deterioration of the patient’s condition. Supplemental oxygen may be required and, sometimes, even intubation [3].

Patient monitoring and assessment can also be performed by chest X-ray. Patients with initial normal chest images may show abnormalities one week after [4]. Initially, patients show no abnormality or a tiny focal alteration on chest X-rays [5]; then, in severe cases of COVID-19, the alteration involves more lobes, becomes bilateral, and consolidation or mixed-patterns occur [6]. Therefore, Brixia score is even more important when successive chest X-ray images are performed, in order to monitor the lung images [7].

Since lung abnormalities in COVID-19 do not always generate a proportional clinical symptomatology, standardizing was required, in an attempt to precisely correlate lung abnormalities on chest X-rays and clinical parameters. Semiquantitative methods to standardize conventional chest X-ray images have been reported [8], each of them partly correlated with patient’s clinical condition [9]. Other scoring methods were required, trying to involve clinical parameters, too.

The reporting of lung alteration on conventional chest X-ray images was so far standardized using Brixia, SARI, RALE and other scores, of which the Brixia score showed the strongest correlation with the clinical condition of the patient [10]. H Brixia score was the best predictor for negative outcomes in the ICU. Ref. [11] The higher the Brixia score, the higher the risk of demise [12].

The Sepsis-3 task force recommends the qSOFA (quick sequential organ failure assessment) score for identifying patients with suspected infection who are at greater risk of poor outcomes. [13,14]. Although the qSOFA score initially aimed to evaluate the complication sequence of disease to generate morbidity, not to predict the outcome, there is a close connection between organ failure and the survival of the patient [15]. The qSOFA score is a very good predictor for COVID-19-generated mortality [16]. COVID-19 patients intubated in the emergency department had a higher qSOFA score and a greater number of pre-existing comorbidities [17].

The study was conducted in accordance with the Declaration of Helsinki, and approved by the Research Ethics Committee of the Elena Doamna Obstetrics and Gynecology University Hospital in Iasi (number 4, 2 April 2020).

The aim of the study was to compare the results of chest X-ray examinations, laboratory tests and the results of the qSOFA scale in the group of patients suffering from COVID-19 infection in the spring and the autumn of 2021. We studied the severity of the course of COVID infection in the Brixia scale compared to qSOFA and biochemical results, in order to check our hypothesis that lung involvement, clinical condition deterioration and blood alterations are more severe in autumn infection compared to spring infection in SARS-CoV-2 critical-infected pregnant patients.

## 2. Materials and Methods

In a prospective study, all pregnant patients admitted at the Elena Doamna Obstetrics and Gynecology Hospital with a critical form of COVID-19 infection between 1 January 2021 and 1 December 2021 were included. Patients who received conventional chest X-rays in another healthcare unit before/during admission in our hospital, where only results were available and not the X-ray images, were excluded from the study. Patients were considered to be at a critical form of COVID-19 when they required admission to the ICU (intensive care unit) due to at least one of the following symptoms: severe dyspnea, oxygen saturation under 95%, extreme fatigue and loss of state of consciousness. All pregnant patients admitted to the hospital were tested by RT-PCR upon admittance, and the positive patients were considered for this study according to the severity of the disease. However, most severe cases of pregnant COVID-19 patients were directed to us from all over the county, because we were a dedicated COVID-19 hospital for obstetrics and gynecology patients; these patients had already taken a positive RT-PCR test and came by ambulance with respiratory support.

Patients underwent conventional chest X-rays with abdomen shielding for fetal protection. A Siemens Polymobil 10 mobile X-ray installation was used. Brixia scores were calculated for every chest image, and the following scores were considered: A score, upon admittance; H and L scores, the highest and lowest scores throughout hospitalization; and E score, the score at the end of hospitalization. For patients who received only one chest examination, the H score was considered. Since we had very few patients in which L score was not the E score or A score, the L Brixia score was eliminated from this study. The Brixia score was calculated by dividing the image of each lung into three zones, from the top of the lung to the base, and each zone was given a number from 0 to 3 as follows: 0—normal image; 1—interstitial opacities; 2—interstitial and alveolar opacities, predominantly interstitial; 3—interstitial and alveolar opacities, predominantly alveolar. The numbers from all quadrants were added, and a final Brixia score ranging from 0 to 18 was obtained [18].

For each day of A, H and E Brixia score, the qSOFA (quick sepsis-related organ failure assessment) score was also calculated as follows: one point was added for each of the following existing issues: systolic blood pressure of 100 mmHg or below, respiratory rate over 22 breaths per minute and Glasgow coma scale under 15 [19]. If any of the previous conditions were present at least once during that specific day, it was considered at qSOFA score calculation.

The blood values on those specific A, H, E Brixia score days, or the closest ones to those days, were considered. We studied the differences between the two groups regarding the Brixia and qSOFA scores, and between the blood values on those specific days: A (admittance), H (highest Brixia score) and E (end of hospitalization).

We divided the patients into two groups: the spring group (group 1, n = 11) from January to May 2021, separated by a two-month period free of severe cases from the autumn group (group 2, n = 7) from August to the end of November 2021. This corresponds to the dominance of the alpha variant of SARS-CoV-2 in spring and the delta variant in autumn of 2021 [20] in Romania. The median age was 32 (27, 37) years old in group 1 and 34 (30, 37) years old in group 2 (*p* = 0.55).

Statistical analysis was performed with SPSS version 18 software (SPSS Inc., Chicago, IL, USA). For descriptive measures, we computed the mean, standard deviation, median and quartiles 1 and 3 (because the variables follow a non-normal distribution). To compare the data, the nonparametric Mann–Whitney U test was applied. The standard significance cut-off at *p* = 0.05 was used to determine our hypothesis conclusion.

## 3. Results

There was one pregnancy with quadruplets in group 1. Most patients delivered in our hospital, but some of them continued their pregnancy (Table 1). In a previous study [21], we showed that there was no significant difference between the two groups regarding maternal and fetal outcomes, but there was a difference regarding the number of days after admittance when patients delivered, which was significantly higher in the autumn group.

### 3.1. Brixia Scores

There was no significant difference between the Brixia scores in the two groups (Table 2), even if the Brixia A score was lower and the Brixia E was higher in the autumn group, meaning that patients were admitted earlier in the ICU (intensive care unit) with less lung involvement and were released home/transferred with higher lung involvement than in the spring group. Symptoms were more dramatic in the autumn group, such that patients were admitted in the ICU with less lung involvement. Patients in the autumn group were released home with a Brixia E score of 11, close to the admittance Brixia A score in the spring group, showing that what was considered concerning lung involvement in the spring variant was less concerning in the autumn group. All the autumn group patients were sent home, with a median Brixia E score of 11, while in the spring group, a similar lung involvement—corresponding to a median Brixia score of 11.33—required admission directly to the ICU. In the spring group, lung involvement tended to decrease between admittance and the end of hospitalization, while in the autumn group, lung involvement continued to increase until reaching H level and then decreased very little until the end of hospitalization. Patients in the autumn group were all sent home, with a median Brixia E score of 11, while at a lower Brixia E score of 9, only one-third of patients in the spring group were sent home; the other two-thirds were sent to other kidney/infectious disease ICU hospitals. This means that the improvement in the clinical aspect of the patient was considered more important than lung involvement; second, the E day chest X-ray was not taken on the very last day of hospitalization, but some days passed from the improvement in the lung involvement to the hospital discharge day. Third, lung involvement on the E day (the end of hospitalization) was worse in autumn patients compared with spring patients, as we initially hypothesized.

### 3.2. qSOFA Scores

The qSOFA scores were not significantly different between the two groups (Table 3); even if the autumn group patients were admitted to the ICU in a better condition (qSOFA A score), their situation (qSOFA score) became worse on H days, and, finally (E day), they were sent home/transferred in a better condition (qSOFA E) than the spring group patients.

This was in accordance with the second part of our hypothesis—that is, in autumn group patients, the deterioration of the clinical condition was worse than in the spring group patients: their qSOFA score increased despite ICU admission, until H day, and then decreased slowly; meanwhile, in the spring group patients, admittance to the ICU favored a slow decrease in qSOFA score.

The qSOFA scores varied inversely compared to lung Brixia scores. In the spring group, lung Brixia scores improved at the end of hospitalization (Brixia E) compared with the highest lung involvement scores (Brixia H); the autumn group had approximately the same lung involvement at the end of hospitalization as in the highest lung involvement, but qSOFA scores showed that the autumn group patients’ overall situation was much better when sent home/transferred (qSOFA E score) compared with those in the spring group, and the final situation (qSOFA E scores) was much better in both groups than on the days of the worst lung involvement (qSOFA H scores).

The Brixia and qSOFA scores also varied unexpectedly in some cases. We present the evolution of the chest X-ray image in a patient, showing that Brixia score and qSOFA score did not always match (Figure 1 and Figure 2).

Lung involvement increased due to evolution of the infection; subsequently, the patient felt worse. Later on, the patient felt better and a chest X-ray was performed, where the lung image showed worsening (Brixia score increased) while the patient’s condition improved (qSOFA score decreased) (Figure 2); only weeks later, their Brixia score came close to normal and the qSOFA score was 0.

This evolution, showing improvement in the patient’s condition (qSOFA decreased) while lung involvement (Brixia score) increased, was seen in two severe-condition patients and only in the spring group (alpha variant); no such evolution was seen in the autumn group (delta variant). Moreover, there were more intermediate images in both patients, where Brixia score increased while qSOFA score decreased and vice versa, with both scores fluctuating in opposite directions until the final stabilization and improvement. When patients felt better, the decreasing qSOFA confirmed the positive evolution, while Brixia score decreased 3–4 days later.

In the autumn group, patients were admitted with a lower Brixia A score, and lung involvement increased quickly during hospitalization. In Figure 3, we present a pregnant patient admitted to the ICU due to severe dyspnea, and the chest X-ray performed that very day showed very small involvement of the left inferior lobe. As she was diagnosed as positive for COVID-19 by RT-PCR and critically ill, she received conventional COVID-19 treatment; three days later, the second X-ray showed increased lung involvement.

Only few patients required intubation, and were only in the spring group (Table 4).

The number of chest X-rays performed varied between patients, according to the clinical state of the patient. There were more chest X-rays performed in the spring group (27) than in the autumn group (12) (Table 5).

### 3.3. Complete Blood Count in the Three Specific Days: Brixia A, H and E

There was no significant difference between the complete-blood-count values of the two groups upon admittance day (Brixia A), except for the red-blood-cell count, which was significantly lower in the autumn group (*p* = 0.01) but still within normal limits. There were differences between the two groups on the H day: neutrophils, red-blood-cell count, hemoglobin and mean corpuscular hemoglobin concentration were lower (*p* = 0.01), while mean corpuscular volume was higher (*p* = 0.02) in the autumn group. There were differences between the two groups on the Brixia E day regarding MID leukocytes (*p* = 0.02), mean corpuscular volume (*p* = 0.02), mean platelet volume (*p* = 0.02) and platelet distribution width (*p* = 0.02). The striking difference was the dramatic decrease in neutrophils, from 6.85 in the spring group to 2.34 (*p* = 0.007) in the autumn group at the end of hospitalization, suggesting that the autumn infection determined a depletion of neutrophils; this is in accordance with our initial hypothesis. On the contrary, the spring variant determined an increase over the normal limit in the number of neutrophils, both on the H day and on the E day; a physiologic increase during infection; as well as a physiologic increase in white-blood-cell count, but only in the spring variant. White-blood-cell count and lymphocyte count remained within normal limits throughout the ICU hospitalization for the autumn group (Table 6).

### 3.4. Coagulation Factors’ Evolution on Three Specific Days: Brixia A, H and E

There was a significant difference only on the Brixia E day of hospitalization, when the APTT value was much higher in the spring group (46.80 versus 28.70; *p* = 0.02), showing that the amount of heparin used to the end of hospitalization was much lower in the autumn group versus the spring group, due to the experience gained throughout the spring period. The APTT value in the spring group was slightly over the normal limit only to the end of hospitalization; later, in the autumn group, APTT was higher than normal only on the worst days (H day) and was slightly higher upon admission. Meanwhile, prothrombin time/activity was slightly higher than normal in both groups upon admission and stayed higher on the H day only in the autumn group (Table 7). This was partly in accordance with our hypothesis, since APTT value depended on the amount of heparin used, and the prothrombin time/activity was not dramatically increased.

### 3.5. Biochemical Blood Values’ Evolution on Three Specific Days: Brixia A, H and E

There was no significant difference between biochemical blood values upon admittance in the two groups, except for calcium values, which were higher in the spring group (spring) (median 9.20 versus 8.80; *p* = 0.02). The CRP values were significantly higher in the autumn group (median 15.00 versus 76.90; *p* = 0.04) on Brixia H day, while the rest of the values were similar in both groups. Only ALT was significantly higher on the Brixia E day in the autumn group compared with the spring group (median 37.00 versus 76.50; *p* = 0.02) (Table 8).

There was a huge increase in liver enzymes (ALT, AST GGT, and LDH) and albumin, reaching higher peaks in the spring group, and remaining high until the end of hospitalization (being sent home/transferring) (Figure 4).

In the autumn group, there was an important increase in GGT value (median 214.05 versus 39.20 in the spring group) at the end of hospitalization. On the contrary, LDH, which was abnormally high in both groups, was much higher in the spring group than in the autumn group during the entire hospitalization and remained high even at the end. Albumin was abnormally high in the spring group on the H and E days, while it was abnormally low on the H day in the autumn group and then returned to normal. The C-reactive protein was also much higher in the autumn patients compared with spring patients (median 76.90 versus 15.00). These values support our hypothesis that the autumn infection generated more aggressive alterations of blood biochemical values than the spring SARS-CoV-2 infection.

## 4. Discussion

If lung involvement, clinical condition and blood alterations are worse in pregnant patients affected by the delta compared with the alpha variant of the SARS-CoV-2 virus, delta patients should be monitored more carefully and the management of follow-up must be adapted.

The clinical condition of the patient can be evaluated using NEWS, Qsofa, SIRS, CRB 65 and other scores, of which qSOFA correlated the least with clinical outcomes [22,23,24,25,26]. We were not interested in the prognosis but in the clinical evaluation on some particular days; therefore, we considered the qSOFA score as the most appropriate to use, because the Brixia and qSOFA scores are the most appropriate methods to describe the severity of COVID-19 [27].

In these severe cases of pregnant COVID-19 patients, the inflammatory response (C-reactive protein) was several times higher in autumn patients (when the delta variant dominated) upon admission and on the Brixia H day, which probably led to the dramatic decrease in neutrophils during the end days of hospitalization in the autumn patients. Meanwhile, the spring infection (when the alpha variant dominated) only triggered an inflammatory response, which increased at a constant level throughout the hospitalization. This is in accordance with Fisman [28] and Rangchaikul [29], who described an increased risk of hospitalization, ICU admission and death due to the delta variant compared with the alpha variant in the general population and in the pregnant population. Furthermore, Eid [30] and Zayet [31] found the delta variant more aggressive and dangerous than the previous variants of concern (alpha included).

The white blood cells and neutrophils slightly and constantly increased in the spring group, but not in the autumn group, where they were within normal limits. No increase or decrease in lymphocyte count was observed in these severe-COVID-19 patients. The pregnancy state probably had a modulatory response and stopped the increase in the white-blood-cell count and lymphocyte count. This would be in accordance with Sievers [32], who reported the antibody response to SARS-CoV-2 infection to be reduced in pregnant women. Our study is in accordance with Areia [33], who found that the SARS-CoV-2 virus determined a decrease in white-blood-cell count in pregnant women; furthermore, no alteration of C-reactive protein or of lymphocytes was found in all infected patients, not only severe cases. In severe cases of pregnant COVID-19 patients, Lasser [34] observed a decreased lymphocyte count; our findings did not corroborate this. Vakili [35] reported leukocytosis and an increased neutrophil ratio in pregnant women, similarly to us. Al-Saadi [36] reported lymphocytopenia and neutrophilia in severe cases of COVID-19, but not specifically in pregnant women. We found neutrophilia but no lymphocytopenia in severe cases of pregnant women. Our findings of increased white-blood-cell count and neutrophil count are supported by the findings of Moghadam [37] in severe cases of pregnant SARS-CoV-2-infected women.

Conventional chest X-ray proved to have an excellent sensitivity in diagnosing lung involvement in moderate-to-severe COVID-19 cases [38], and the Brixia scale proved to be the best method to evaluate conventional X-ray chest images during the COVID-19 pandemics [10]. Maroldi [39] correlated the A, L and E Brixia scores with the outcomes of COVID-19 pneumonia patients, which makes sense: the lower the Brixia score, the less the lungs were affected upon admission (A score); at the end of hospitalization (E score); and, of course, at the lowest lung involvement (L score). The less the lungs are involved, the better the outcome. Maroldi [39] also correlated the increased Brixia H score (the largest and worst lung involvement) with the increased probability of negative outcomes and death of patients, which also makes sense. Henley [40] and Palsencia-Martinez [41] also correlated the highest Brixia scores in a modified version and second Brixia score, respectively, with the probability of later need for ventilation or intubation in COVID-19 patients. However, they did not study the clinical state of the patient together with the Brixia scores, nor did they compare alpha with delta patients as we did. Henley [40] also reported a mean Brixia score upon admittance to the ICU of 9, and the need for ventilation starting with an 11.5 Brixia score. We reported a median Brixia score of 11.3 upon admittance to the ICU in the spring patients, and 8 in the autumn patients, showing that admittance to the ICU was very late in the spring patients, who stayed home as long as they could. Conversely, in autumn, COVID-19 patients went to hospital earlier due to the more intense delta symptoms and perhaps due to the abundant media information advising citizens to consult a physician as soon as they felt symptoms and tested positive for the SARS-CoV-2 antigen. Borghesi [42] also correlated the Brixia H score with the outcomes in COVID-19 patients, showing that patients with an H score under 8 had a good prognosis. We cannot confirm that, because our median admittance (A) Brixia score was over 11.3 in spring and over 8 in autumn patients; from there, it went higher to the Brixia H (highest) score. Balbi [43], d’Souza [44] and Au-Yong [45] correlated the Brixia A score with outcomes in COVID-19 patients. Setiawati [46] found that severe cases of COVID-19 pneumonia had a Brixia score over 6, Gatti [47] over 7 upon admission and Boari [48] over 8, but the first and last authors did not specify which Brixia score they used: A, H or E. Similarly, Hoang [49] and Gurtoo [50] also correlated Brixia score with the outcomes in COVID-19 patients but did not specify which Brixia score they used in calculations—either A, H or E score—or if there was more than one chest X-ray taken of each patient. In another study, Boari [51] even correlated the Brixia score with the lung involvement at the posthospitalization follow-up. However, none of them correlated Brixia score with a specific variant of concern, alpha or delta, nor did they describe pregnant patients as we did.

We used the qSOFA score to assess the clinical state of patients on every specific Brixia day—A, H and E—and compared the evolution. In spring patients, the qSOFA score was high upon admittance (A day), decreased slowly until the H day, and then decreased significantly toward the end of hospitalization. In autumn patients, the qSOFA score continued to increase from A day to H day, then decreased dramatically to the end (E day). Moreover, the qSOFA score on admittance (A) day was higher in the spring infection compared with the autumn infection, meaning that the patients’ clinical state was worse in spring patients than in autumn patients upon admittance. During hospitalization, clinical status improved quickly in spring patients (qSOFA score decreased) but continued to deteriorate in autumn patients (qSOFA score increased) towards the H day.

As described above, there was no correlation between the Brixia score and the qSOFA (*p* = 1) score on each particular day—A, H or E—in either the spring or the autumn group; the Brixia score (radiological evaluation of lung involvement) increased or decreased days later compared to the more reliable qSOFA score, which exactly described the clinical status of the patient that day. However, neither Brixia score nor qSOFA score in one particular day could predict the radiological or clinical evolution of the patient during the next days, in either spring or autumn patients. This is in accordance with Cagino [52], Heldt [53] and Alencar [54], who found no correlation between qSOFA score and outcomes in pregnant or nonpregnant COVID-19 patients. Still, Aashik [55] found the qSOFA score to predict the mortality of COVID-19 patients, while Vikas [56] found the Brixia A score and qSOFA score to predict outcomes in severe forms of pregnant COVID-19 patients.

There was no correlation between either Brixia score or qSOFA score on any particular day (A, H or E) and the blood coagulation factors or biochemical blood values (*p* = 1), in any group. There were no major alterations of the blood cell indices in the two groups, except for the dramatic difference in neutrophils, which was 6.85 in the spring group versus 2.34 (*p* = 0.007) in the autumn group at the end of hospitalization, suggesting that the autumn infection determined a depletion of neutrophils. In the spring group, APTT was slightly over the limit only on the end day, at the same level with the admission day in the autumn group; APTT continued to increase up to the H day in this second group and then decreased, showing that hypercoagulability induced by the autumn infection required more heparin and/or patients were already on heparin when transferred to the ICU.

Since the viral loading of SARS-CoV-2 variants was the most important in the lungs, and second in the liver [57], blood liver enzymes’ level increased. Jang [58] reported that AST elevated to 126 IU/L and LDH elevated to 409 IU/mL in delta variant pneumonia patients; we observed lower AST and higher LDH in our autumn pregnant patients, when delta variant dominated. No comparisons between alpha- and delta-variant-infected pregnant patients regarding coagulation factors, liver enzymes or blood biochemistry have been found so far.

In this study, we demonstrated our hypothesis that severe cases of COVID-19 patients behaved worse in autumn 2021, when the delta variant dominated, compared with the spring patients, when the alpha variant dominated, with regard to lung involvement, coagulation factors and blood enzymes.

Since the delta variant is much more contagious than the alpha variant, even in a fully vaccinated population [59,60], the risk of viral outbreak is still present. Knowing the unpredictable evolution of lung involvement, blood components and coagulation factors will help health professionals react quicker, even in severe cases, especially in the vulnerable population of pregnant persons.

The study has several weaknesses. First, the number of critical-form patients included in each group is small, since our COVID-19-dedicated ICU is small, with few beds, even when they doubled during the pandemics. Second, the moment of performing a conventional chest X-ray may be chosen differently in other hospitals and their results may be different. Therefore, larger studies with more numerous participants are required to confirm these results.

## 5. Conclusions

The evolution of autumn 2021 SARS-CoV-2 infection in severe forms in pregnant patients was worse than that of the spring 2021 patients, with regard to lung involvement, clinical evolution and alteration of different blood components. Knowing the unpredictable change in these values will help health professionals react quicker to the severe forms of pregnant COVID-19 patients.

## Figures and Tables

**Figure 1 healthcare-10-01423-f001:**
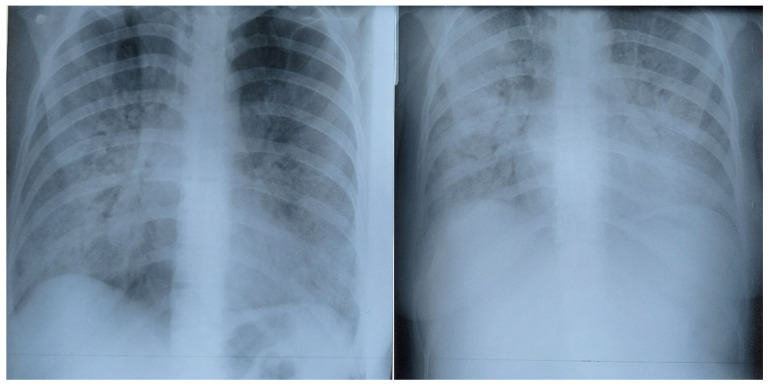
Pregnant patient in the ICU. On the left, Brixia A score 12, qSOFA A score 0. On the right, two days later, Brixia score 14, qSOFA score 1 (respiratory rate over 22 breaths per minute). Patient in the spring group.

**Figure 2 healthcare-10-01423-f002:**
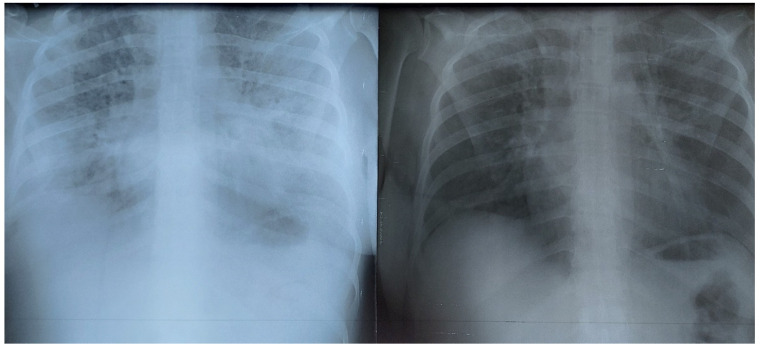
On the left, four days after the last image above, Brixia H score 15, Qsofa H score 0. On the right, one week later, Brixia E score 2, qSOFA E score 0. Same patient in the spring group.

**Figure 3 healthcare-10-01423-f003:**
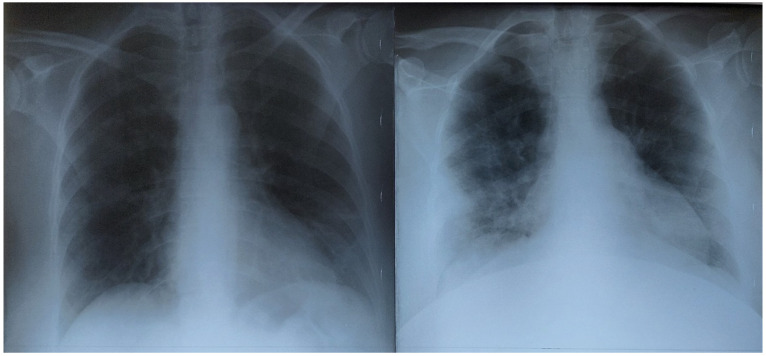
On the left, upon admission, Brixia A score 1, qSOFA score 1. On the right, three days later, Brixia H score 8, qSOFA score 1 (respiratory rate over 22 breaths per minute). Patient in the autumn group.

**Figure 4 healthcare-10-01423-f004:**
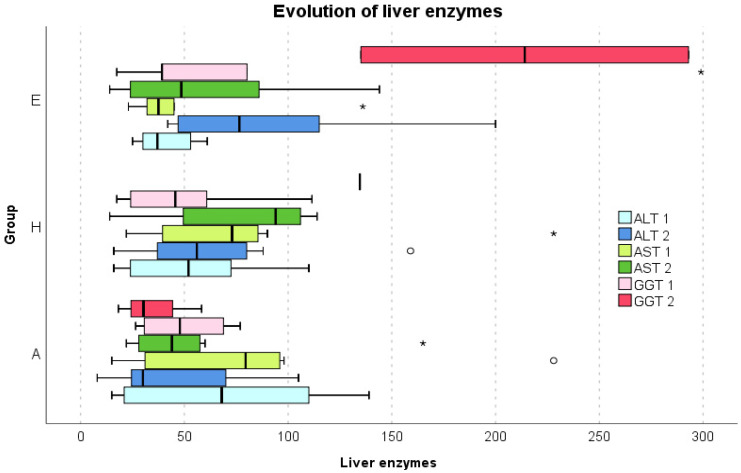
Evolution of some liver enzymes in the two groups on the three Brixia days: A, H and E. All values are above the normal limits. ° = outlier value, * = extreme outlier value, | = median value. A—admittance day (Brixia A), H—the day of highest lung involvement of COVID-19 in the ICU (Brixia H), E—the end day of hospitalization in the ICU (Brixia E), ALT—alanine aminotransferase, AST—aspartate aminotransferase, GGT—gamma glutamyl traspeptidase, 1—spring group, 2—autumn group.

**Table 1 healthcare-10-01423-t001:** Description of the pregnant population in the two groups: mean values.

Patients	Spring Group (n = 11)	Autumn Group (n = 7)	*p*-Value
Gestational age upon admission (weeks)	31.36	29.85	0.412
Gestational age upon delivery (weeks)	32.33	34.33	0.353
Singletons	8/9 (88.88%)	3/3 (100%)	0.287

**Table 2 healthcare-10-01423-t002:** Brixia scores in the two groups: median values and interquartile range.

Score	Spring Group	Autumn Group	*p*
Brixia A	13 (7,15)	8 (1,15)	0.61
Brixia H	13 (8,16)	12 (11,13)	0.81
Brixia E	9 (6,13)	11 (11,11)	0.502

A—admittance, H—highest score, E—end of hospitalization.

**Table 3 healthcare-10-01423-t003:** qSOFA scores in the two groups: median values and interquartile range.

Score	Spring Group	Autumn Group	Significance, *p*
qSOFA A	1 (0,2)	0 (0,2)	0.31
qSOFA H	1 (0,1)	1 (0,3)	0.32
qSOFA E	0 (0,1)	0 (0,0)	0.18

A—admittance, H—highest Brixia score day, E—end of hospitalization.

**Table 4 healthcare-10-01423-t004:** Patients who required intubation in the two groups, and Brixia score (mean value) when intubation was required.

Patients Requiring Intubation	Spring Group (n = 11)	Autumn Group (n = 7)
Number of patients requiring intubation	2	0
Brixia score when intubation was required	11.5	0

**Table 5 healthcare-10-01423-t005:** Number of chest X-rays performed per patient.

Number of X-Rays Performed per Patient	Spring Group (n = 11)	Autumn Group (n = 7)
1 (one)	5 (45.45%)	5 (71.42%)
2 (two)	3 (27.27%)	0 (0%)
3 (three)	0 (0%)	1 (14.28%)
4 (four)	2 (18.18%)	1 (14.28%)
8 (eight)	1 (9.09%)	0 (0%)

**Table 6 healthcare-10-01423-t006:** Abnormal and normal values of complete blood count in the two groups on three specific days in the ICU—Brixia A, H and E: median values and interquartile range.

Parameters.	Group	A Day	H Day	E Day	Normal Range
WBC	Spring	8.14 (5.52; 12.05)	11.2 (9.07; 15.28)	13.53 (8.27; 15.22)	4.5–11
	Autumn	6.75 (6.46; 9.81)	8.87 (6.46; 10.61)	10.39 (8.74; 12.67)	
NEUT	Spring	4.27 (3.79; 9.12)	7.00 (5.67; 8.10)	6.85 (4.91; 8.29)	2.5–6
	Autumn	3.71 (2.62; 4.05)	3.14 (1.82; 3.90)	2.34 (1.32; 4.12)	
LYM	Spring	1.11 (0.93; 1.75)	1.65 (1.22; 2.34)	2.07 (1.82; 2.68)	1–4
	Autumn	1.11 (0.96; 1.77)	1.48 (0.96; 2.71)	2.51 (1.90; 2.96)	
PDW	Spring	16.36 (14.02; 18.21)	17.35 (15.37; 18.86)	17.57 (15.90; 18.82)	10–17.9%
	Autumn	18.41 (17.53; 19.10)	18.15 (17.68; 19.47)	20.95 (19.17; 21.71)	
PCT	Spring	0.19 (0.13; 0.21)	0.20 (0.19; 0.23)	0.27 (0.21; 0.30)	0.20–0.36%
	Autumn	0.16 (0.13; 0.17)	0.17 (0.13; 0.24)	0.27 (0.18; 0.27)	

A—admittance day (Brixia A), H—the day of highest lung involvement of COVID-19 in the ICU (Brixia H), E—the end day of hospitalization in the ICU (Brixia E), PDW—platelet-derived width, PCT—plateletcrit, WBC—white-blood-cell count, NEUT—neutrophils. Normal values are replaced by “-”.

**Table 7 healthcare-10-01423-t007:** Abnormal and normal values of coagulation factors’ count in the two groups on three specific days in the ICU—Brixia A, H and E: median values and interquartile range.

Parameters	Group	A Day	H Day	E Day	Normal Range
APTT	Spring	36.30 (34; 41.20)	37.10 (29.70; 47.20)	46.80 (42.10; 47.20)	4.5–11
	Autumn	42.80 (41.50; 48.20)	42.80 (39.10; 56.70)	29.70 (27.10; 36.40)	
Prothrombin activity	Spring	90.40 (104.50; 124.20)	97.10 (95.80; 107.70)	98.60 (91.80; 103.00)	2.5–6
	Autumn	107.10 (80.60; 122.80)	132.80 (72.70; 148.30)	85.80 (71.30; 102.30)	
Prothrombin time	Spring	11.70 (11.30; 13.50)	12.50 (11.80; 12.60)	12.40 (12.10; 12.90)	11–12.5
	Autumn	12.40 (11.30; 13.50)	11.30 (11.10; 14.00)	13.65 (12.80; 14.65)	

A—admittance day (Brixia A), H—the day of highest lung involvement of COVID-19 in the ICU (Brixia H), E—the end day of hospitalization in the ICU (Brixia E), APTT—activated partial thromboplastin time. Normal values are replaced by “-”.

**Table 8 healthcare-10-01423-t008:** Abnormal and normal biochemical blood values in the two groups on three specific days in the ICU—Brixia A, H and E: median values and interquartile range.

Parameters	Group	A Day	H Day	E Day	Normal Range
ALT	Spring	68.00 (21.00; 110.00)	52.00 (24.00; 72.50)	37.00 (30.00; 53.00)	19–25 UI/L
	Autumn	30.00 (21.00; 94.00)	56.00 (28.00; 88.00)	76.50 (47.00; 115.00)	
AST	Spring	79.50 (31.00; 96.00)	73.00 (39.50; 85.50)	37.50 (32.00; 45.00)	5–40 UI/L
	Autumn	44.00 (24.00; 60.00)	94.00 (44.00; 109.00)	48.50 (24.00; 86.00)	
LDH	Spring	800.00 (513.00; 816.00)	722.00 (620.00; 943.00)	870.50 (665.00; 1420.00)	140–280 U/L
	Autumn	334.00 (334.00; 334.00)	715.00 (535.00; 895.00)	511.00 (454.00; 778.00)	
CRP	Spring	12.35 (6.38; 38.30)	15.00 (6.20; 18.00)	32.78 (2.15; 38.57)	<6
	Autumn	88.10 (87.90; 101.90)	76.90 (15.80; 122.30)	22.30 (10.60; 42.30)	
GGT	Spring	47.85 (30.70; 68.85)	45.60 (24.10; 60.80)	39.20 (39.10; 80.20)	0–30 U/L
	Autumn	30.30 (18.20; 58.30)	134.60 (134.60; 134.60)	214.05 (135.10; 293.00)	
Albumin	Spring	-	17.15 (2.91; 31.38)	28.27 (23.75; 32.79)	3.5–5 g/dL
	Autumn	-	2.44 (2.44; 2.44)	3.31 (3.26;3.47)	
Total proteins	Spring	-	5.44 (5.14; 5.63)	5.31 (4.92; 6.85)	6–8.3 g/dL
	Autumn	-	5.67 (5.61; 5.73)	5.91 (5.51; 6.00)	
Iron	Spring	46.80 (34.90; 210.60)	100.80 (64.30;102.90)	102.90 (64.30;110.70)	37–145 μg/dL
	Autumn	29.20 (11.00; 47.40)	32.95 (11.00; 54.90)	164.90 (164.90; 164.90)	
Alkaline Reserve	Spring	-	-	21.60 (17.65; 28.35)	22–29 mmol/L
	Autumn	-	-	19.50 (19.50; 19.50)	

A—admittance day (Brixia A), H—the day of highest lung involvement of COVID-19 in the ICU (Brixia H), E—the end day of hospitalization in the ICU (Brixia E), NEUT—neutrophils, APTT—activated partial thromboplastin time, ALT—alanine aminotransferase, AST—aspartate aminotransferase, LDH—lactate dehydrogenase, CRP—C-reactive protein, GGT—gamma glutamyl transpeptidase. Normal values are replaced by “-”.

## Data Availability

All data are available from the corresponding author upon reasonable request.

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
