# Peer review of "Brixia and qSOFA Scores, Coagulation Factors and Blood Values in Spring versus Autumn 2021 Infection in Pregnant Critical COVID-19 Patients: A Preliminary Study"

_healthcare, 2022, doi:10.3390/healthcare10081423_

Round 1
Reviewer 1 Report
Thank you for the oportunity to review your paper. It is well and clearly written.
I have one major remark: as it deals with pregnant women, I would expect a detailed description of the pregnant population. (gestational week, singletons, twins, outcome of pregnancy etc.) I did not find any of these informations.
Also the results could be more pecific in context of pregnancy lenth and number of offspring (if there were any multiples).
Author Response
Response to Reviewer 1
Open Review
(x) I would not like to sign my review report
( ) I would like to sign my review report
English language and style
( ) Extensive editing of English language and style required
( ) Moderate English changes required
( ) English language and style are fine/minor spell check required
(x) I don't feel qualified to judge about the English language and style
|
Yes |
Can be improved |
Must be improved |
Not applicable |
|
|
Does the introduction provide sufficient background and include all relevant references? |
( ) |
( ) |
(x) |
( ) |
|
Are all the cited references relevant to the research? |
(x) |
( ) |
( ) |
( ) |
|
Is the research design appropriate? |
(x) |
( ) |
( ) |
( ) |
|
Are the methods adequately described? |
( ) |
( ) |
(x) |
( ) |
|
Are the results clearly presented? |
(x) |
( ) |
( ) |
( ) |
|
Are the conclusions supported by the results? |
(x) |
( ) |
( ) |
( ) |
Comments and Suggestions for Authors
Thank you for the oportunity to review your paper. It is well and clearly written.
I have one major remark: as it deals with pregnant women, I would expect a detailed description of the pregnant population. (gestational week, singletons, twins, outcome of pregnancy etc.) I did not find any of these informations. Also the results could be more pecific in context of pregnancy lenth and number of offspring (if there were any multiples).
-We added, as highlighted in RED, in Results:
„3. Results
Table 1. Description of the pregnant population in the two groups: mean values.
|
Patients |
Group 1 (n = 11) |
Group 2 (n = 7) |
P-value |
|
Gestational age upon admission (weeks) |
31.36 |
29.85 |
0.412 |
|
Gestational age upon delivery (weeks) |
32.33 |
34.33 |
0.353 |
|
Singletons |
8/9 (88.88%) |
3/3 (100%) |
0.287 |
There was one pregnancy with quadruplets in group 1. Most patients delivered in our hospital, but some of them continued their pregnancy. In a previous study [11] we showed that there was no significant difference between the two groups as regarded maternal and fetal outcomes, but there was a difference regarding the number of days after admittance when patients delivered, which was significantly higher in group 2.”
And in References:
- Covali, R.; Socolov, D.; Pavaleanu, I.; Akad, M.; Boiculese, L.V.; Socolov, R. Brixia Score in Outcomes of Alpha versus Delta Variant of Infection in Pregnant Critical COVID-19 Patients. Int. J. Transl. Med.2022, 2, 66-77. https://doi.org/10.3390/ijtm2010007”
Submission Date
18 June 2022
Date of this review
12 Jul 2022 15:44:40

Reviewer 2 Report
The work comparing patients suffering from COVID-19 infection in the spring and autumn of 2021 compares the results of X-ray tests, laboratory tests and the results of the qSOFA scale. The work is very interesting, but requires major revision.
1. Introduction:
· The introduction is inadequate to the topic of the thesis. I propose to describe more about the use of X-ray and Brixia scale in covid infection than the use of CT.
· The SOFA scale paragraph should be in discussion. In the introduction, it is better to describe the utility of the SOFA scale.
· The last paragraph of the introduction should also be included in the discussion.
· The introduction should contain information about the consent of the bioethics committee to conduct the research.
· At the end of the introduction, there should be a clearly defined goal of the study.
· The aim of the study is probably to compare the results of X-ray examinations, laboratory tests and the results of the qSOFA scale in the group of patients suffering from COVID-19 infection in the spring and the autumn of 2021. It would be interesting to compare the severity of the course of Covid infection in the Brixia scale compared to SOFA and biochemical results.
2. Material and methods:
· There is no characteristic of the research group in the materials and methods.
· There is no information on how many patients had only one X-ray performed.
3. Results:
• All tables should present the results as median and interquartile range for non-normal distributions.
• if you compare the spring and fall disease groups, the groups should be called spring and fall, there is no evidence that the patients suffered from the alpha or delta variants. These are only suspicions.
• Table 3 presents different parameters for the period A, H, E. Moreover, PDW, PCT is not clinically significant. I suggest rewriting the table to present WBC (for A, H, E), neutrophils (for A, H, E), lymphocytes (for A, H, E).
• In table 4, I suggest modifying the table to present the parameters for A, H, E.
• Table 5 shows the different parameters for the period A, H, E. I suggest that you rewrite the table as above
• There is a lack of information on the clinical condition of the patients, whether they required intubation and at what BRIXIA scale score.
Author Response
Response to Reviewer 2
Open Review
( ) I would not like to sign my review report
(x) I would like to sign my review report
English language and style
( ) Extensive editing of English language and style required
( ) Moderate English changes required
( ) English language and style are fine/minor spell check required
(x) I don't feel qualified to judge about the English language and style
|
Yes |
Can be improved |
Must be improved |
Not applicable |
|
|
Does the introduction provide sufficient background and include all relevant references? |
( ) |
( ) |
(x) |
( ) |
|
Are all the cited references relevant to the research? |
(x) |
( ) |
( ) |
( ) |
|
Is the research design appropriate? |
( ) |
( ) |
(x) |
( ) |
|
Are the methods adequately described? |
( ) |
( ) |
(x) |
( ) |
|
Are the results clearly presented? |
( ) |
( ) |
(x) |
( ) |
|
Are the conclusions supported by the results? |
( ) |
( ) |
(x) |
( ) |
Comments and Suggestions for Authors
The work comparing patients suffering from COVID-19 infection in the spring and autumn of 2021 compares the results of X-ray tests, laboratory tests and the results of the qSOFA scale. The work is very interesting, but requires major revision.
- Introduction:
- The introduction is inadequate to the topic of the thesis. I propose to describe more about the use of X-ray and Brixia scale in covid infection than the use of CT.
-We added, as highlighted in YELLOW:
“Mild and moderate cases of COVID-19 may show no abnormality on chest X-rays, while in severe cases opacities are visible in both lungs, either interstitial and/or alveolar type opacities. These opacities are situated to the periphery of the lungs, in lower lobes, and have a tendency to unite. With proper therapy, opacities slowly disappear, but sometimes they involve the majority of lung area with catastrophic deterioration of the patient’s condition. Supplemental oxygen may be required and, sometimes, even intubation. [3]
Patient monitoring and assessment can also be performed by chest X-ray. Patients with initial normal chest images may show abnormalities one week after [4]. Initially, patients show no abnormality or a tiny focal alteration on chest X-rays [5], then, in severe cases of COVID-19, the alteration involves more lobes, becomes bilateral, and consolidation, or mixed-patterns occur [6]. Therefore, Brixia score is even more important when successive chest X-ray images are performed, in order to monitor the lung images. [7]
Since lung abnormalities in COVID-19 do not always generate a proportional clinical symptomatology, standardizing was required, in an attempt to precisely correlate lung abnormalities on chest X-rays and clinical parameters. Semiquantitative methods to standardize conventional chest X-rays images have been reported [8], each of them partly correlated with patient’s clinical condition [9]. Other scoring methods were required, trying to involve clinical parameters, too.
Reporting of lung alteration on conventional chest X-ray images was so far standardized using Brixia, SARI, RALE and other scores, of which the Brixia score showed the strongest correlation with the clinical condition of the patient [10]. H Brixia score was the best predictor for negative outcomes in the ICU. [11]. The higher the Brixia score, the higher the risk of demise [12]. “
- The SOFA scale paragraph should be in discussion.
-We moved it in Discussion, as highlighted in YELLOW:
“Discussion
If lung involvement, clinical condition and blood alterations are worse in pregnant patients affected by the delta compared with the alpha variant of the SARS-CoV-2 virus, delta patients should be monitored more carefully and management of follow-up must be adapted.
The clinical condition of the patient can be evaluated using NEWS, Qsofa, SIRS, CRB 65 and other scores, of which qSOFA correlated the least with clinical outcomes [4-6]. We were not interested in the prognosis but in the clinical evaluation on some particular days; therefore, we considered qSOFA score as the most appropriate to use, because the Brixia and qSOFA scores are the most appropriate methods to describe the severity of COVID-19 [7].”
-In the introduction, it is better to describe the utility of the SOFA scale.
-We added, as highlighted in YELLOW:
“The Sepsis-3 task force recommends the qSOFA (quick sequential organ failure assessment) score for identifying patients with suspected infection who are at greater risk of poor outcomes. [13,14]. Though the qSOFA score initially aimed to evaluate the complication sequence of disease to generate morbidity, not to predict the outcome, there is a close connection between organ failure and survival of the patient [15]. The qSOFA score is a very good predictor for the COVID-19 generated mortality [16]. COVID-19 patients intubated in the emergency department had a higher qSOFA score and a greater number of pre-existing comorbidities. [17]”
- The last paragraph of the introduction should also be included in the discussion.
-We did move it in Discussion, as highlighted in YELLOW:
- Discussion
If lung involvement, clinical condition and blood alterations are worse in pregnant patients affected by the delta compared with the alpha variant of the SARS-CoV-2 virus, delta patients should be monitored more carefully and management of follow-up must be adapted.
- The introduction should contain information about the consent of the bioethics committee to conduct the research.
-We added, to the end of Introduction:
The study was conducted in accordance with the Declaration of Helsinki, and approved by the Research Ethics Committee of the Elena Doamna Obstetrics and Gynecology University Hospital in Iasi (number 4, 2 April 2020).
- At the end of the introduction, there should be a clearly defined goal of the study.
- The aim of the study is probably to compare the results of X-ray examinations, laboratory tests and the results of the qSOFA scale in the group of patients suffering from COVID-19 infection in the spring and the autumn of 2021. It would be interesting to compare the severity of the course of Covid infection in the Brixia scale compared to SOFA and biochemical results.
-We added, in the end of Introduction, as you suggested:
The aim of the study was to compare the results of chest X-ray examinations, laboratory tests and the results of the qSOFA scale in the group of patients suffering from COVID-19 infection in the spring and the autumn of 2021. We studied the severity of the course of Covid infection in the Brixia scale compared to SOFA and biochemical results.
- Material and methods:
- There is no characteristic of the research group in the materials and methods.
-We added, highlighted in RED:
- Results
Table 1. Description of the pregnant population in the two groups: mean values.
|
Patients |
Group 1 (n = 11) |
Group 2 (n = 7) |
P-value |
|
Gestational age upon admission (weeks) |
31.36 |
29.85 |
0.412 |
|
Gestational age upon delivery (weeks) |
32.33 |
34.33 |
0.353 |
|
Singletons |
8/9 (88.88%) |
3/3 (100%) |
0.287 |
There was one pregnancy with quadruplets in group 1. Most patients delivered in our hospital, but some of them continued their pregnancy. In a previous study [11] we showed that there was no significant difference between the two groups as regarded maternal and fetal outcomes, but there was a difference regarding the number of days after admittance when patients delivered, which was significantly higher in group 2.
- There is no information on how many patients had only one X-ray performed.
-We added, as highlighted in YELLOW:
“The number of chest X-rays performed varied between patients, according to the clinical state of the patient. There were more chest X-rays performed in the spring group (27) than in the fall group (12). (Table 5).
Table 5. Number of chest X-rays performed per patient
|
Number of X-rays performed per patient |
Group 1 (n = 11) |
Group 2 (n = 7) |
|
1 (one) |
5 (45.45%) |
5 (71.42%) |
|
2 (two) |
3 (27.27%) |
0 (0%) |
|
3 (three) |
0 (0%) |
1 (14.28%) |
|
4 (four) |
2 (18.18%) |
1 (14.28%) |
|
8 (eight) |
1 (9.09%) |
0 (0%) |
- Results:
- All tables should present the results as median and interquartile range for non-normal distributions.
-We did, as highlighted in YELLOW:
Table 2. Brixia scores in the two groups: median values and interquartile range.
|
Score |
Spring group |
Fall group |
P |
|
Brixia A |
13 (7,15) |
8 (1,15) |
0.61 |
|
Brixia H |
13 (8,16) |
12 (11,13) |
0.81 |
|
Brixia E |
9 (6,13) |
11 (11,11) |
0.502 |
Table 3. qSOFA scores in the two groups: median values and interquartile range.
|
Score |
Spring group |
Fall group |
Significance, P |
|
qSOFA A |
1 (0,2) |
0 (0,2) |
0.31 |
|
qSOFA H |
1 (0,1) |
1 (0,3) |
0.32 |
|
qSOFA E |
0 (0,1) |
0 (0,0) |
0.18 |
Table 6. Abnormal values of complete blood count in the two groups on three specific days in the ICU—Brixia A, H and E: median values and interquartile range.
|
Count |
Spring group |
Fall group |
Normal range |
|
A PDW |
16.36 (14.02;19.21) |
18.41 (17.53;19.10) |
10-17.9% |
|
A PCT |
0.19 (0.13; 0.21) |
0.16 (0.13; 0.17) |
0.20-0.36% |
|
H WBC |
11.20 (9.07; 15.28) |
8.87(6.46;10.61) |
4.5-11 |
|
H NEUT |
7.00 (5.67; 8.10) |
3.14 (1.82;3.90) |
2.5-6 |
|
H PDW |
17.35 (15.39;18.86) |
18.15 (17.68; 19.47) |
10-17.9% |
|
E WBC |
13.53 (8.27; 15.22) |
10.39 (8.74;12.67) |
4.5-11 |
|
E NEUT |
6.85 (4.91; 8.29) |
2.34 (1.32;4.12) |
2.5-6 |
|
E PDW |
17.57 (15.90;18.82) |
20.95 (19.17; 21.71) |
10-17.9% |
Table 7. Abnormal values of coagulation factors count in the two groups on three specific days in the ICU—Brixia A, H and E: median values and interquartile range.
|
Count |
Spring group |
Fall group |
Normal range |
|
A APTT |
36.30 (34;41.20) |
42.80 (41.50;48.20) |
30-40s |
|
A Prothrombin activity |
09.40 (104.50; 124.20) |
107.10 (80.60; 122.80) |
85-100% |
|
H APTT |
37.10 (29.70;47.20) |
42.80 (39.10; 56.70) |
30-40s |
|
H Prothrombin activity |
97.10 (95.80;107.70) |
132.80 (72.70; 148.30) |
85-100% |
|
E APTT |
46.80 (42.10; 47.20) |
29.70 (27.10;36.40) |
30-40s |
|
E Prothrombin time |
12.40 (12.10;12.90) |
13.65 (12.80; 14.65) |
11-12.5 |
Table 8. Abnormal biochemical blood values in the two groups on three specific days in the ICU—Brixia A, H and E: median values and interquartile range.
|
Count |
Spring group |
Fall group |
Normal range |
|
A ALT |
68.00 (21.00; 110.00) |
30.00 (21.00; 94.00) |
19-25UI/L |
|
A AST |
79.50 (31.00; 96.00) |
44.00 (24.00; 60.00) |
5-40UI/L |
|
A LDH |
800.00 (513.00; 816.00) |
334.00 (334.00; 334.00) |
140-280U/L |
|
A CRP |
12.35 (6.38; 38.30) |
88.10 (87.90; 101.90) |
<6 |
|
A Direct Bilirubin |
0.38 (0.21; 0.62) |
0.23 (0.18;0.28) |
<0.30mg/dL |
|
A GGT |
47.85 (30.70; 68.85) |
30.30 (18.20; 58.30) |
0-30U/L |
|
A Triglycerides |
254.60 (177.15; 453.10) |
233.80 (233.80; 233.80) |
<150mg/dL |
|
A Iron |
46.80 (34.90; 210.60) |
29.20 (11.00; 47.40) |
37-145mg/dL |
|
H Albumin |
17.15 (2.91; 31.38) |
2.44 (2.44; 2.44) |
3.5-5g/dL |
|
H ALT |
52.00 (24.00; 72.50) |
56.00 (28.00; 88.00) |
19-25UI/L |
|
H AST |
73.00 (39.50; 85.50) |
94.00 (44.00; 109.00) |
5-40UI/L |
|
H Total proteins |
5.44 (5.14; 5.63) |
5.67 (5.61; 5.73) |
6-8.3mg/dL |
|
H LDH |
722.00 (620.00; 943.00) |
715.00 (535.00; 895.00) |
140-280U/L |
|
H CRP |
15.00 (6.20; 18.00) |
76.90 (15.80; 122.30) |
<6 |
|
H Direct Bilirubin |
0.28 (0.15; 0.61) |
0.34 (0.29; 0.39) |
<0.3mg/dL |
|
H GGT |
45.60 (24.10; 60.80) |
134.60 (134.60; 134.60) |
0-30U/L |
|
H Na |
132.50 (129.90;139.70) |
146.70 (142.80; 150.60) |
135-145mEq/L |
|
H Iron |
100.80 (64.30;102.90) |
32.95 (11.00; 54.90) |
37-145μg/dL |
|
E Albumin |
28.27 (23.75; 32.79) |
3.31 (3.26;3.47) |
3.5-5g/dL |
|
E ALT |
37.00 (30.00; 53.00) |
76.50 (47.00; 115.00) |
19-25UI/L |
|
E AST |
37.50 (32.00; 45.00) |
48.50 (24.00; 86.00) |
5-40UI/L |
|
E Total proteins |
5.31 (4.92; 6.85) |
5.91 (5.51; 6.00) |
6-8.3g/dL |
|
E LDH |
870.50 (665.00; 1420.00) |
511.00 (454.00; 778.00) |
140-280U/L |
|
E Glycemia |
81.10 (66.00;102.80) |
105.35 (85.80; 118.00) |
<100mg/dL |
|
E CRP |
32.78 (2.15; 38.57) |
22.30 (10.60; 42.30) |
<6 |
|
E GGT |
39.20 (39.10; 80.20) |
214.05 (135.10; 293.00) |
0-30U/L |
|
E Cholesterol |
271.75 (211.00; 342.10) |
462.00 (462.00; 462.00) |
<200mg/dL |
|
E Triglycerides |
306.30 (253.00; 309.80) |
258.70 (258.70; 258.70) |
<150mg/dL |
|
E Iron |
102.90 (64.30;110.70) |
164.90 (164.90; 164.90) |
37-145μg/dL |
|
E Alkaline Reserve |
21.60 (17.65; 28.35) |
19.50 (19.50; 19.50) |
22-29mmol/L |
- if you compare the spring and fall disease groups, the groups should be called spring and fall, there is no evidence that the patients suffered from the alpha or delta variants. These are only suspicions.
-We changed, as highlighted in YELLOW:
Table 2. Brixia scores in the two groups: mean and median values.
|
Score |
Spring group |
Fall group |
P |
|
Brixia A |
11.33±4.76 13 (7,15) |
8.00±9.90 8 (1,15) |
0.61 |
|
Brixia H |
11.09±5.09 13 (8,16) |
11.57±2.82 12 (11,13) |
0.81 |
|
Brixia E |
9.67±4.23 9 (6,13) |
11.00±.00 11 (11,11) |
0.502 |
Table 3. qSOFA scores in the two groups: mean and median values.
|
Score |
Spring group |
Fall group |
Significance, P |
|
qSOFA A |
1.18±0.98 1 (0,2) |
0.71±0.95 0 (0,2) |
0.31 |
|
qSOFA H |
0.73±0.90 1 (0,1) |
1.29±1.25 1 (0,3) |
0.32 |
|
qSOFA E |
0.45±0.52 0 (0,1) |
0.14±0.38 0 (0,0) |
0.18 |
Table 4. Abnormal values of complete blood count in the two groups on three specific days in the ICU—Brixia A, H and E: mean and median values.
|
Count |
Spring group |
Fall group |
Normal range |
|
A PDW |
16.35±3.26 16.36 (14.02;19.21) |
18.05±1.34 18.41 (17.53;19.10) |
10-17.9% |
|
A PCT |
0.18±0.05 0.19 (0.13; 0.21) |
0.15±0.03 0.16 (0.13; 0.17) |
0.20-0.36% |
|
H WBC |
12.43±4.25 11.20 (9.07; 15.28) |
9.09±2.75 8.87(6.46;10.61) |
4.5-11 |
|
H NEUT |
7.39±2.35 7.00 (5.67; 8.10) |
3.02±1.39 3.14 (1.82;3.90) |
2.5-6 |
|
H PDW |
17.12±2.12 17.35 (15.39;18.86) |
18.76±1.70 18.15 (17.68; 19.47) |
10-17.9% |
|
E WBC |
12.97±5.23 13.53 (8.27; 15.22) |
10.62±2.04 10.39 (8.74;12.67) |
4.5-11 |
|
E NEUT |
7.50±4.10 6.85 (4.91; 8.29) |
2.78±1.77 2.34 (1.32;4.12) |
2.5-6 |
|
E PDW |
17.79±2.51 17.57 (15.90;18.82) |
20.71±1.61 20.95 (19.17; 21.71) |
10-17.9% |
Table 5. Abnormal values of coagulation factors count in the two groups on three specific days in the ICU—Brixia A, H and E: mean and median values.
|
Count |
Spring group |
Fall group |
Normal range |
|
A APTT |
39.30±7.11 36.30 (34;41.20) |
43.80±5.69 42.80 (41.50;48.20) |
30-40s |
|
A Prothrombin activity |
109.00±17.38 109.40 (104.50; 124.20) |
102.73±25.27 107.10 (80.60; 122.80) |
85-100% |
|
H APTT |
37.39±7.55 37.10 (29.70;47.20) |
61.80±55.30 42.80 (39.10; 56.70) |
30-40s |
|
H Prothrombin activity |
101.99±21.77 97.10 (95.80;107.70) |
117.73±44.25 132.80 (72.70; 148.30) |
85-100% |
|
E APTT |
44.26±4.99 46.80 (42.10; 47.20) |
30.73±4.97 29.70 (27.10;36.40) |
30-40s |
|
E Prothrombin time |
12.54±0.67 12.40 (12.10;12.90) |
13.73±1.20 13.65 (12.80; 14.65) |
11-12.5 |
Table 6. Abnormal biochemical blood values in the two groups on three specific days in the ICU—Brixia A, H and E: mean and median values.
|
Count |
Spring group |
Fall group |
Normal range |
|
A ALT |
68.60±47.02 68.00 (21.00; 110.00) |
47.43±37.46 30.00 (21.00; 94.00) |
19-25UI/L |
|
A AST |
79.70±60.66 79.50 (31.00; 96.00) |
57.43±49.64 44.00 (24.00; 60.00) |
5-40UI/L |
|
A LDH |
969.141±131.49 800.00 (513.00; 816.00) |
334.00±0 334.00 (334.00; 334.00) |
140-280U/L |
|
A CRP |
24.73±27.71 12.35 (6.38; 38.30) |
84.10±47.90 88.10 (87.90; 101.90) |
<6 |
|
A Direct Bilirubin |
0.39±0.24 0.38 (0.21; 0.62) |
0.23±0.05 0.23 (0.18;0.28) |
<0.30mg/dL |
|
A GGT |
49.78±23.24 47.85 (30.70; 68.85) |
35.60±20.57 30.30 (18.20; 58.30) |
0-30U/L |
|
A Triglycerides |
315.13±196.64 254.60 (177.15; 453.10) |
233.80±0 233.80 (233.80; 233.80) |
<150mg/dL |
|
A Iron |
122.75±163.13 46.80 (34.90; 210.60) |
29.20±25.74 29.20 (11.00; 47.40) |
37-145mg/dL |
|
H Albumin |
17.15±20.13 17.15 (2.91; 31.38) |
2.44±0 2.44 (2.44; 2.44) |
3.5-5g/dL |
|
H ALT |
52.88±32.32 52.00 (24.00; 72.50) |
66.43±47.63 56.00 (28.00; 88.00) |
19-25UI/L |
|
H AST |
80.75±64.20 73.00 (39.50; 85.50) |
76.14±38.50 94.00 (44.00; 109.00) |
5-40UI/L |
|
H Total proteins |
5.36±0.33 5.44 (5.14; 5.63) |
5.67±0.08 5.67 (5.61; 5.73) |
6-8.3mg/dL |
|
H LDH |
1109.33±1048.79 722.00 (620.00; 943.00) |
715.00±254.56 715.00 (535.00; 895.00) |
140-280U/L |
|
H CRP |
20.30±21.14 15.00 (6.20; 18.00) |
71.44±49.39 76.90 (15.80; 122.30) |
<6 |
|
H Direct Bilirubin |
0.33±0.24 0.28 (0.15; 0.61) |
0.34±0.07 0.34 (0.29; 0.39) |
<0.3mg/dL |
|
H GGT |
50.83±33.73 45.60 (24.10; 60.80) |
134.60±0 134.60 (134.60; 134.60) |
0-30U/L |
|
H Na |
130.84±16.06 132.50 (129.90;139.70) |
146.70±5.52 146.70 (142.80; 150.60) |
135-145mEq/L |
|
H Iron |
130.62±136.56 100.80 (64.30;102.90) |
32.95±31.04 32.95 (11.00; 54.90) |
37-145μg/dL |
|
E Albumin |
28.27±6.39 28.27 (23.75; 32.79) |
3.37±0.18 3.31 (3.26;3.47) |
3.5-5g/dL |
|
E ALT |
40.50±13.81 37.00 (30.00; 53.00) |
92.83±59.17 76.50 (47.00; 115.00) |
19-25UI/L |
|
E AST |
51.83±41.90 37.50 (32.00; 45.00) |
60.83±47.88 48.50 (24.00; 86.00) |
5-40UI/L |
|
E Total proteins |
5.92±1.45 5.31 (4.92; 6.85) |
5.90±1.12 5.91 (5.51; 6.00) |
6-8.3g/dL |
|
E LDH |
1065.67±519.60 870.50 (665.00; 1420.00) |
594.00±185.44 511.00 (454.00; 778.00) |
140-280U/L |
|
E Glycemia |
82±20.80 81.10 (66.00;102.80) |
105.18±19.78 105.35 (85.80; 118.00) |
<100mg/dL |
|
E CRP |
31.49±28.95 32.78 (2.15; 38.57) |
25.07±16.03 22.30 (10.60; 42.30) |
<6 |
|
E GGT |
94.98±116.29 39.20 (39.10; 80.20) |
214.05±111.65 214.05 (135.10; 293.00) |
0-30U/L |
|
E Cholesterol |
278.08±90.82 271.75 (211.00; 342.10) |
462.00±0 462.00 (462.00; 462.00) |
<200mg/dL |
|
E Triglycerides |
293.62±84.21 306.30 (253.00; 309.80) |
258.70±0 258.70 (258.70; 258.70) |
<150mg/dL |
|
E Iron |
96.74±35.32 102.90 (64.30;110.70) |
164.90±0 164.90 (164.90; 164.90) |
37-145μg/dL |
|
E Alkaline Reserve |
23.00±76.63 21.60 (17.65; 28.35) |
19.50±0 19.50 (19.50; 19.50) |
22-29mmol/L |
- Table 3 presents different parameters for the period A, H, E. Moreover, PDW, PCT is not clinically significant. I suggest rewriting the table to present WBC (for A, H, E), neutrophils (for A, H, E), lymphocytes (for A, H, E).
-We did replace Table 3 (now Table 6):
|
Count |
Spring group |
Fall group |
Normal range |
|
A PDW |
16.36 (14.02;19.21) |
18.41 (17.53;19.10) |
10-17.9% |
|
A PCT |
0.19 (0.13; 0.21) |
0.16 (0.13; 0.17) |
0.20-0.36% |
|
H WBC |
11.20 (9.07; 15.28) |
8.87(6.46;10.61) |
4.5-11 |
|
H NEUT |
7.00 (5.67; 8.10) |
3.14 (1.82;3.90) |
2.5-6 |
|
H PDW |
17.35 (15.39;18.86) |
18.15 (17.68; 19.47) |
10-17.9% |
|
E WBC |
13.53 (8.27; 15.22) |
10.39 (8.74;12.67) |
4.5-11 |
|
E NEUT |
6.85 (4.91; 8.29) |
2.34 (1.32;4.12) |
2.5-6 |
|
E PDW |
17.57 (15.90;18.82) |
20.95 (19.17; 21.71) |
10-17.9% |
With:
Table 6. Abnormal and normal values of complete blood count in the two groups on three specific days in the ICU—Brixia A, H and E: median values and interquartile range.
|
Parameters |
Group |
A day |
H day |
E day |
Normal range |
|
WBC |
Spring |
8.14 (5.52;12.05) |
11.2 (9.07;15.28) |
13.53 (8.27;15.22) |
4.5-11 |
|
Fall |
6.75 (6.46;9.81) |
8.87 (6.46;10.61) |
10.39 (8.74;12.67) |
||
|
NEUT |
Spring |
4.27 (3.79;9.12) |
7.00 (5.67;8.10) |
6.85 (4.91;8.29) |
2.5-6 |
|
Fall |
3.71 (2.62;4.05) |
3.14 (1.82;3.90) |
2.34 (1.32;4.12) |
||
|
LYM |
Spring |
1.11 (.93;1.75) |
1.65 (1.22;2.34) |
2.07 (1.82;2.68) |
1-4 |
|
Fall |
1.11 (.96;1.77) |
1.48 (.96;2.71) |
2.51 (1.90;2.96) |
||
|
PDW |
Spring |
16.36 (14.02;18.21) |
17.35 (15.37;18.86) |
17.57 (15.90;18.82) |
10-17.9% |
|
Fall |
18.41 (17.53;19.10) |
18.15 (17.68;19.47) |
20.95 (19.17;21.71) |
||
|
PCT |
Spring |
.19 (.13;.21) |
.20 (.19;.23) |
.27 (.21;.30) |
0.20-0.36% |
|
Fall |
.16 (.13;.17) |
.17 (.13;.24) |
.27 (.18;.27) |
- In table 4, I suggest modifying the table to present the parameters for A, H, E.
-We did replace Table 4 (now Table 7):
Table 7. Abnormal values of coagulation factors count in the two groups on three specific days in the ICU—Brixia A, H and E: median values and interquartile range.
|
Count |
Spring group |
Fall group |
Normal range |
|
A APTT |
36.30 (34;41.20) |
42.80 (41.50;48.20) |
30-40s |
|
A Prothrombin activity |
09.40 (104.50; 124.20) |
107.10 (80.60; 122.80) |
85-100% |
|
H APTT |
37.10 (29.70;47.20) |
42.80 (39.10; 56.70) |
30-40s |
|
H Prothrombin activity |
97.10 (95.80;107.70) |
132.80 (72.70; 148.30) |
85-100% |
|
E APTT |
46.80 (42.10; 47.20) |
29.70 (27.10;36.40) |
30-40s |
|
E Prothrombin time |
12.40 (12.10;12.90) |
13.65 (12.80; 14.65) |
11-12.5 |
With:
Table 7. Abnormal and normal values of coagulation factors count in the two groups on three specific days in the ICU—Brixia A, H and E: median values and interquartile range.
|
Parameters |
Group |
A day |
H day |
E day |
Normal range |
|
APTT |
Spring |
36.30 (34;41.20) |
37.10 (29.70;47.20) |
46.80 (42.10; 47.20) |
4.5-11 |
|
Fall |
42.80 (41.50;48.20) |
42.80 (39.10; 56.70) |
29.70 (27.10;36.40) |
||
|
Prothrombin activity |
Spring |
90.40 (104.50; 124.20) |
97.10 (95.80;107.70) |
98.60 (91.80;103.00) |
2.5-6 |
|
Fall |
107.10 (80.60; 122.80) |
132.80 (72.70; 148.30) |
85.80 (71.30;102.30) |
||
|
Prothrombin time |
Spring |
11.70 (11.30;13.50) |
12.50 (11.80;12.60) |
12.40 (12.10;12.90) |
11-12.5 |
|
Fall |
12.40 (11.30;13.50) |
11.30 (11.10;14.00) |
13.65 (12.80;14.65) |
- Table 5 shows the different parameters for the period A, H, E. I suggest that you rewrite the table as above
-We rewrote Table 5 (now Table 8):
Table 8. Abnormal biochemical blood values in the two groups on three specific days in the ICU—Brixia A, H and E: median values and interquartile range.
|
Count |
Spring group |
Fall group |
Normal range |
|
A ALT |
68.00 (21.00; 110.00) |
30.00 (21.00; 94.00) |
19-25UI/L |
|
A AST |
79.50 (31.00; 96.00) |
44.00 (24.00; 60.00) |
5-40UI/L |
|
A LDH |
800.00 (513.00; 816.00) |
334.00 (334.00; 334.00) |
140-280U/L |
|
A CRP |
12.35 (6.38; 38.30) |
88.10 (87.90; 101.90) |
<6 |
|
A Direct Bilirubin |
0.38 (0.21; 0.62) |
0.23 (0.18;0.28) |
<0.30mg/dL |
|
A GGT |
47.85 (30.70; 68.85) |
30.30 (18.20; 58.30) |
0-30U/L |
|
A Triglycerides |
254.60 (177.15; 453.10) |
233.80 (233.80; 233.80) |
<150mg/dL |
|
A Iron |
46.80 (34.90; 210.60) |
29.20 (11.00; 47.40) |
37-145mg/dL |
|
H Albumin |
17.15 (2.91; 31.38) |
2.44 (2.44; 2.44) |
3.5-5g/dL |
|
H ALT |
52.00 (24.00; 72.50) |
56.00 (28.00; 88.00) |
19-25UI/L |
|
H AST |
73.00 (39.50; 85.50) |
94.00 (44.00; 109.00) |
5-40UI/L |
|
H Total proteins |
5.44 (5.14; 5.63) |
5.67 (5.61; 5.73) |
6-8.3mg/dL |
|
H LDH |
722.00 (620.00; 943.00) |
715.00 (535.00; 895.00) |
140-280U/L |
|
H CRP |
15.00 (6.20; 18.00) |
76.90 (15.80; 122.30) |
<6 |
|
H Direct Bilirubin |
0.28 (0.15; 0.61) |
0.34 (0.29; 0.39) |
<0.3mg/dL |
|
H GGT |
45.60 (24.10; 60.80) |
134.60 (134.60; 134.60) |
0-30U/L |
|
H Na |
132.50 (129.90;139.70) |
146.70 (142.80; 150.60) |
135-145mEq/L |
|
H Iron |
100.80 (64.30;102.90) |
32.95 (11.00; 54.90) |
37-145μg/dL |
|
E Albumin |
28.27 (23.75; 32.79) |
3.31 (3.26;3.47) |
3.5-5g/dL |
|
E ALT |
37.00 (30.00; 53.00) |
76.50 (47.00; 115.00) |
19-25UI/L |
|
E AST |
37.50 (32.00; 45.00) |
48.50 (24.00; 86.00) |
5-40UI/L |
|
E Total proteins |
5.31 (4.92; 6.85) |
5.91 (5.51; 6.00) |
6-8.3g/dL |
|
E LDH |
870.50 (665.00; 1420.00) |
511.00 (454.00; 778.00) |
140-280U/L |
|
E Glycemia |
81.10 (66.00;102.80) |
105.35 (85.80; 118.00) |
<100mg/dL |
|
E CRP |
32.78 (2.15; 38.57) |
22.30 (10.60; 42.30) |
<6 |
|
E GGT |
39.20 (39.10; 80.20) |
214.05 (135.10; 293.00) |
0-30U/L |
|
E Cholesterol |
271.75 (211.00; 342.10) |
462.00 (462.00; 462.00) |
<200mg/dL |
|
E Triglycerides |
306.30 (253.00; 309.80) |
258.70 (258.70; 258.70) |
<150mg/dL |
|
E Iron |
102.90 (64.30;110.70) |
164.90 (164.90; 164.90) |
37-145μg/dL |
|
E Alkaline Reserve |
21.60 (17.65; 28.35) |
19.50 (19.50; 19.50) |
22-29mmol/L |
And we replaced it with:
Table 8. Abnormal and normal biochemical blood values in the two groups on three specific days in the ICU—Brixia A, H and E: median values and interquartile range.
|
Parameters |
Group |
A day |
H day |
E day |
Normal range |
|
ALT |
Spring |
68.00 (21.00; 110.00) |
52.00 (24.00; 72.50) |
37.00 (30.00; 53.00) |
19-25UI/L |
|
Fall |
30.00 (21.00; 94.00) |
56.00 (28.00; 88.00) |
76.50 (47.00; 115.00) |
||
|
AST |
Spring |
79.50 (31.00; 96.00) |
73.00 (39.50; 85.50) |
37.50 (32.00; 45.00) |
5-40UI/L |
|
Fall |
44.00 (24.00; 60.00) |
94.00 (44.00; 109.00) |
48.50 (24.00; 86.00) |
||
|
LDH |
Spring |
800.00 (513.00; 816.00) |
722.00 (620.00; 943.00) |
870.50 (665.00; 1420.00) |
140-280U/L |
|
Fall |
334.00 (334.00; 334.00) |
715.00 (535.00; 895.00) |
511.00 (454.00; 778.00) |
||
|
CRP |
Spring |
12.35 (6.38; 38.30) |
15.00 (6.20; 18.00) |
32.78 (2.15; 38.57) |
<6 |
|
Fall |
88.10 (87.90; 101.90) |
76.90 (15.80; 122.30) |
22.30 (10.60; 42.30) |
||
|
GGT |
Spring |
47.85 (30.70; 68.85) |
45.60 (24.10; 60.80) |
39.20 (39.10; 80.20) |
0-30U/L |
|
Fall |
30.30 (18.20; 58.30) |
134.60 (134.60; 134.60) |
214.05 (135.10; 293.00) |
||
|
Albumin |
Spring |
- |
17.15 (2.91; 31.38) |
28.27 (23.75; 32.79) |
3.5-5g/dL |
|
Fall |
- |
2.44 (2.44; 2.44) |
3.31 (3.26;3.47) |
||
|
Total proteins |
Spring |
- |
5.44 (5.14; 5.63) |
5.31 (4.92; 6.85) |
6-8.3g/dL |
|
Fall |
- |
5.67 (5.61; 5.73) |
5.91 (5.51; 6.00) |
||
|
Iron |
Spring |
46.80 (34.90; 210.60) |
100.80 (64.30;102.90) |
102.90 (64.30;110.70) |
37-145μg/dL |
|
Fall |
29.20 (11.00; 47.40) |
32.95 (11.00; 54.90) |
164.90 (164.90; 164.90) |
||
|
Alkaline Reserve |
Spring |
- |
- |
21.60 (17.65; 28.35) |
22-29mmol/L |
|
Fall |
- |
- |
19.50 (19.50; 19.50) |
- There is a lack of information on the clinical condition of the patients, whether they required intubation and at what BRIXIA scale score.
-We added, highlighted in YELLOW:
Table 4. Patients who required intubation in the two groups, and Brixia score (mean value) when intubation was required
|
Patients requiring intubation |
Group 1 (n = 11) |
Group 2 (n = 7) |
|
Number of patients requiring intubation |
2 |
0 |
|
Brixia score when intubation was required |
11.5 |
0 |
Submission Date
18 June 2022
Date of this review
13 Jul 2022 14:15:21

Round 2
Reviewer 1 Report
thank you for accepting reviewers comments.
Author Response
Response 2 to Reviewer 1
Open Review
(x) I would not like to sign my review report
( ) I would like to sign my review report
English language and style
( ) Extensive editing of English language and style required
( ) Moderate English changes required
(x) English language and style are fine/minor spell check required
( ) I don't feel qualified to judge about the English language and style
|
Yes |
Can be improved |
Must be improved |
Not applicable |
|
|
Does the introduction provide sufficient background and include all relevant references? |
(x) |
( ) |
( ) |
( ) |
|
Are all the cited references relevant to the research? |
(x) |
( ) |
( ) |
( ) |
|
Is the research design appropriate? |
(x) |
( ) |
( ) |
( ) |
|
Are the methods adequately described? |
(x) |
( ) |
( ) |
( ) |
|
Are the results clearly presented? |
(x) |
( ) |
( ) |
( ) |
|
Are the conclusions supported by the results? |
(x) |
( ) |
( ) |
( ) |
Comments and Suggestions for Authors
thank you for accepting reviewers comments.
-Thank you for your kind advices!
Submission Date
18 June 2022
Date of this review
22 Jul 2022 09:49:38

Reviewer 2 Report
The work is very interesting, thank you for the improvements made. The article requires a few minor corrections.
1. Please read the entire article (and title) and consistently describe the groups as spring and autumn, of course you can point out that it is highly probable that the spring group had alpha and the autumn group had delta.
2. In the article you refer to” your hypothesis”, but after the modifications of the aim of the study there is no the sentence with this hypothesis. Please read the article carefully so that it is logically consistent with the corrections made.
3. I propose to quote:
Krawczyk P, Jastrzebska A, Szczeklik W, Andres J. Perioperative lung ultrasound pattern changes in patients undergoing gynecological procedures - a prospective observational study. Ginekol Pol. 2021;92(4):271-277. doi: 10.5603/GP.a2021.0017.
Krawczyk P, Jaśkiewicz R, Huras H, Kołak M. Obstetric Anesthesia Practice in the Tertiary Care Center: A 7-Year Retrospective Study and the Impact of the COVID-19 Pandemic on Obstetric Anesthesia Practice. J Clin Med. 2022 Jun 2;11(11):3183.
Author Response
Response 2 to Reviewer 2
Open Review
( ) I would not like to sign my review report
(x) I would like to sign my review report
English language and style
( ) Extensive editing of English language and style required
( ) Moderate English changes required
( ) English language and style are fine/minor spell check required
(x) I don't feel qualified to judge about the English language and style
|
Yes |
Can be improved |
Must be improved |
Not applicable |
|
|
Does the introduction provide sufficient background and include all relevant references? |
(x) |
( ) |
( ) |
( ) |
|
Are all the cited references relevant to the research? |
(x) |
( ) |
( ) |
( ) |
|
Is the research design appropriate? |
(x) |
( ) |
( ) |
( ) |
|
Are the methods adequately described? |
(x) |
( ) |
( ) |
( ) |
|
Are the results clearly presented? |
(x) |
( ) |
( ) |
( ) |
|
Are the conclusions supported by the results? |
( ) |
(x) |
( ) |
( ) |
Comments and Suggestions for Authors
The work is very interesting, thank you for the improvements made. The article requires a few minor corrections.
- Please read the entire article (and title) and consistently describe the groups as spring and autumn, of course you can point out that it is highly probable that the spring group had alpha and the autumn group had delta.
-We did describe them as spring and autumn groups, including title change, as highlighted in YELLOW:
-Title:
Brixia and qSOFA scores, coagulation factors and blood values in spring versus autumn 2021 infection in pregnant critical COVID-19 patients
-Abstract:
Abstract: (1) Background: From the recent variants of concern of the SARS-CoV-2 virus, in which the delta variant generated more negative outcomes than the alpha, we hypothesized that lung involvement, clinical condition deterioration and blood alterations were also more severe in autumn infection, when the delta variant dominated, compared with spring infections, when the alpha variant dominated, in severely infected pregnant patients. (2) Methods: In a prospective study, all pregnant patients admitted to the ICU of the Elena Doamna Obstetrics and Gynecology Hospital with a critical form of COVID-19 infection—spring group (n=11) and autumn group (n=7)—between 1 January 2021 and 1 December 2021 were included. Brixia scores were calculated for every patient: A score, upon admittance; H score, the highest score throughout hospitalization; and E score, at the end of hospitalization. For each day of Brixia A, H or E score, the qSOFA (quick sepsis-related organ failure assessment) score was calculated, and the blood values were also considered. (3) Results: Brixia E score, C-reactive protein, GGT and LDH were much higher, while neutrophil count was much lower in autumn compared with spring critical-form pregnant patients. (4) Conclusions: the autumn infection generated more dramatic alterations than spring infection in pregnant patients with critical forms of COVID-19.
-In Introduction:
- Introduction
Of the recent variants of concern of the SARS-CoV-2 virus, two of them were devastating all around the world: the alpha variant, with a much higher transmissibility than previous ones [1], and the delta variant, which affected even fully vaccinated persons [2]. The delta variant generated more negative outcomes than the alpha in the general population and in the pregnant population. We hypothesize that lung involvement, clinical condition deterioration and blood alterations are also more severe in autumn infection, when the delta variant dominated, compared with spring infections, when the alpha variant dominated, in SARS-CoV-2 critical-infected pregnant patients.
-in Results:
- Results
Table 1. Description of the pregnant population in the two groups: mean values.
|
Patients |
Spring group (n = 11) |
Autumn group (n = 7) |
P-value |
|
Gestational age upon admission (weeks) |
31.36 |
29.85 |
0.412 |
|
Gestational age upon delivery (weeks) |
32.33 |
34.33 |
0.353 |
|
Singletons |
8/9 (88.88%) |
3/3 (100%) |
0.287 |
There was one pregnancy with quadruplets in group 1. Most patients delivered in our hospital, but some of them continued their pregnancy. In a previous study [21] we showed that there was no significant difference between the two groups as regarded maternal and fetal outcomes, but there was a difference regarding the number of days after admittance when patients delivered, which was significantly higher in autumn group.
3.1. Brixia scores
There was no significant difference between the Brixia scores in the two groups (Table 2), even if the Brixia A score was lower and the Brixia E was higher in the autumn group, meaning that patients were admitted earlier in the ICU (intensive care unit) with less lung involvement and were released home/transferred with higher lung involvement than in the spring group. Symptoms were more dramatic in the autumn group, such that patients were admitted in the ICU with less lung involvement. Patients in the autumn group were released home with a Brixia E score of 11, close to the admittance Brixia A score in the spring group, showing that what was considered concerning lung involvement in the spring variant was less concerning in the autumn group. All the autumn group patients were released home, with a median Brixia E score of 11, while in the spring group, a similar lung involvement—corresponding to a median Brixia score of 11.33—required admission directly in the ICU. In the spring group, lung involvement tended to decrease between admittance and the end of hospitalization, while in the autumn group, lung involvement continued to increase until reaching H level and then decreased very little until the end of hospitalization. Patients in the autumn group were all released home, with a median Brixia E score of 11, while at a lower Brixia E score of 9, only one-third of patients in the spring group were sent home, the other two-thirds were sent to other kidney/infectious disease ICU hospitals. This means that the improvement in the clinical aspect of the patient was considered more important than lung involvement; second, the E day chest X-ray was not taken on the very last day of hospitalization, but some days passed from the improvement of the lung involvement to the hospital discharge day. Third, lung involvement on the E day (the end of hospitalization) was worse in autumn patients compared with spring patients, as we initially hypothesized.
Table 2. Brixia scores in the two groups: median values and interquartile range.
|
Score |
Spring group |
Autumn group |
P |
|
Brixia A |
13 (7,15) |
8 (1,15) |
0.61 |
|
Brixia H |
13 (8,16) |
12 (11,13) |
0.81 |
|
Brixia E |
9 (6,13) |
11 (11,11) |
0.502 |
A—admittance, H—highest score, E—end of hospitalization.
3.2. qSOFA scores
The qSOFA scores were not significantly different between the two groups (Table 3); even if autumn group patients were admitted to the ICU in a better condition (qSOFA A score), their situation (qSOFA score) became worse on H days, and finally (E day), they were released home/transferred in a better condition (qSOFA E) than the spring group patients.
Table 3. qSOFA scores in the two groups: median values and interquartile range.
|
Score |
Spring group |
Autumn group |
Significance, P |
|
qSOFA A |
1 (0,2) |
0 (0,2) |
0.31 |
|
qSOFA H |
1 (0,1) |
1 (0,3) |
0.32 |
|
qSOFA E |
0 (0,1) |
0 (0,0) |
0.18 |
A—admittance, H—highest Brixia score day, E—end of hospitalization.
This was in accordance with the second part of our hypothesis—that is, in autumn group patients, deterioration of the clinical condition was worse than in the spring group patients: their qSOFA score increased despite ICU admission, until H day, and then decreased slowly; meanwhile, in the spring group patients, admittance to the ICU favored slow decrease of qSOFA score.
The qSOFA scores varied inversely compared to lung Brixia scores. In the spring group, lung Brixia scores improved at the end of hospitalization (Brixia E) compared with the highest lung involvement scores (Brixia H); the autumn group had approximately the same lung involvement at the end of hospitalization as in the highest lung involvement, but qSOFA scores showed that the autumn group patients’ overall situation was much better when released home/transferred (qSOFA E score) compared with those in the spring group, and the final situation (qSOFA E scores) was much better in both groups than on the days of the worst lung involvement (qSOFA H scores).
The Brixia and qSOFA scores also varied unexpectedly in some cases. We present the evolution of the chest X-ray image in a patient, showing that Brixia score and qSOFA score did not always match (Figures 1 and 2).
Figure 1. Pregnant patient in the ICU. On the left, Brixia A score 12, qSOFA A score 0. On the right, two days later, Brixia score 14, qSOFA score 1 (respiratory rate over 22 breaths per minute). Patient in the spring group.
Lung involvement increased due to evolution of the infection; subsequently, the patient felt worse. Later on, the patient felt better and a chest X-ray was performed, where the lung image showed worsening (Brixia score increased) while the patient’s condition improved (qSOFA score decreased) (Figure 2); only weeks later, their Brixia score came close to normal and qSOFA score was 0.
Figure 2. On the left, four days after the last image above, Brixia H score 15, Qsofa H score 0. On the right, one week later, Brixia E score 2, qSOFA E score 0. Same patient in the spring group.
This evolution, showing improvement in the patient’s condition (qSOFA decreased) while lung involvement (Brixia score) increased, was seen in two severe-condition patients and only in the spring group (alpha variant); no such evolution was seen in the autumn group (delta variant). Moreover, there were more intermediate images in both patients, where Brexia score increased while qSOFA score decreased and vice versa, with both scores fluctuating in opposite directions until the final stabilization and improvement. When patients felt better, the decreasing qSOFA confirmed the positive evolution, while Brixia score decreased 3-4 days later.
In the autumn group, patients were admitted with a lower Brixia A score, and lung involvement increased quickly during hospitalization. In Figure 3, we present a pregnant patient admitted to the ICU due to severe dyspneea, and the chest X-ray performed that very day showed very small involvement of the left inferior lobe. As she was diagnosed as positive for COVID-19 by RT-PCR and critically ill, she received conventional COVID-19 treatment; three days later, the second X-ray showed increased lung involvement.
Figure 3. On the left, upon admission, Brixia A score 1, qSOFA score 1. On the right, three days later, Brixia H score 8, qSOFA score 1 (respiratory rate over 22 breaths per minute). Patient in the autumn group.
Only few patients required intubation, and only in spring group (Table 4).
Table 4. Patients who required intubation in the two groups, and Brixia score (mean value) when intubation was required
|
Patients requiring intubation |
Spring group (n = 11) |
Autumn group (n = 7) |
|
Number of patients requiring intubation |
2 |
0 |
|
Brixia score when intubation was required |
11.5 |
0 |
The number of chest X-rays performed varied between patients, according to the clinical state of the patient. There were more chest X-rays performed in the spring group (27) than in the autumn group (12). (Table 5).
Table 5. Number of chest X-rays performed per patient
|
Number of X-rays performed per patient |
Spring group (n = 11) |
Autumn group (n = 7) |
|
1 (one) |
5 (45.45%) |
5 (71.42%) |
|
2 (two) |
3 (27.27%) |
0 (0%) |
|
3 (three) |
0 (0%) |
1 (14.28%) |
|
4 (four) |
2 (18.18%) |
1 (14.28%) |
|
8 (eight) |
1 (9.09%) |
0 (0%) |
3.3. Complete blood count in the three specific days: Brixia A, H and E
There was no significant difference between the complete blood count values of the two groups upon admittance day (Brixia A), except for the red blood cell count, which was significantly lower in the autumn group (P=0.01) but still within normal limits. There were differences between the two groups on the H day: neutrophils, red blood cell count, hemoglobin and mean corpuscular hemoglobin concentration were lower (P=0.01), while mean corpuscular volume was higher (P=0.02) in the autumn group. There were differences between the two groups on the Brixia E day regarding MID leukocytes (P=0.02), mean corpuscular volume (P=0.02), mean platelet volume (P=0.02) and platelet distribution width (P=0.02). The striking difference was the dramatic decrease in neutrophils, from 6.85 in the spring group to 2.34 (P=0.007) in the autumn group at the end of hospitalization, suggesting that the autumn infection determined a depletion of neutrophils; this is in accordance with our initial hypothesis. On the contrary, the spring variant determined an increase over the normal limit in the number of neutrophils, both on the H day and on the E day; a physiologic increase during infection; as well as a physiologic increase in white blood cell count, but only in the spring variant. White blood cell count and lymphocyte count remained within normal limits throughout the ICU hospitalization for the autumn group (Table 6).
Table 6. Abnormal and normal values of complete blood count in the two groups on three specific days in the ICU—Brixia A, H and E: median values and interquartile range.
|
Parameters |
Group |
A day |
H day |
E day |
Normal range |
|
WBC |
Spring |
8.14 (5.52;12.05) |
11.2 (9.07;15.28) |
13.53 (8.27;15.22) |
4.5-11 |
|
Autumn |
6.75 (6.46;9.81) |
8.87 (6.46;10.61) |
10.39 (8.74;12.67) |
||
|
NEUT |
Spring |
4.27 (3.79;9.12) |
7.00 (5.67;8.10) |
6.85 (4.91;8.29) |
2.5-6 |
|
Autumn |
3.71 (2.62;4.05) |
3.14 (1.82;3.90) |
2.34 (1.32;4.12) |
||
|
LYM |
Spring |
1.11 (.93;1.75) |
1.65 (1.22;2.34) |
2.07 (1.82;2.68) |
1-4 |
|
Autumn |
1.11 (.96;1.77) |
1.48 (.96;2.71) |
2.51 (1.90;2.96) |
||
|
PDW |
Spring |
16.36 (14.02;18.21) |
17.35 (15.37;18.86) |
17.57 (15.90;18.82) |
10-17.9% |
|
Autumn |
18.41 (17.53;19.10) |
18.15 (17.68;19.47) |
20.95 (19.17;21.71) |
||
|
PCT |
Spring |
.19 (.13;.21) |
.20 (.19;.23) |
.27 (.21;.30) |
0.20-0.36% |
|
Autumn |
.16 (.13;.17) |
.17 (.13;.24) |
.27 (.18;.27) |
A—admittance day (Brixia A), H—the day of highest lung involvement of COVID-19 in the ICU (Brixia H), E—the end day of hospitalization in the ICU (Brixia E), PDW—platelet-derived width, PCT—plateletcrit, WBC—white blood cell count, NEUT—neutrophils. Normal values are replaced by “-“.
3.4. Coagulation factors evolution on three specific days: Brixia A, H and E.
There was a significant difference only on the Brixia E day of hospitalization, when the APTT value was much higher in the spring group (46.80 versus 28.70; P=0.02), showing that the amount of heparin used to the end of hospitalization was much lower in the autumn group versus the spring group, due to the experience gained throughout the spring period. The APTT value in the spring group was slightly over the normal limit only to the end of hospitalization; later, in the autumn group, APTT was higher than normal only on the worst days (H day) and was slightly higher upon admission. Meanwhile, prothrombin time/activity was slightly higher than normal in both groups upon admission and stayed higher on the H day only in the autumn group (Table 7). This was partly in accordance with our hypothesis, since APTT value depended on the amount of heparin used, and the prothrombin time/activity were not dramatically increased.
Table 7. Abnormal and normal values of coagulation factors count in the two groups on three specific days in the ICU—Brixia A, H and E: median values and interquartile range.
|
Parameters |
Group |
A day |
H day |
E day |
Normal range |
|
APTT |
Spring |
36.30 (34;41.20) |
37.10 (29.70;47.20) |
46.80 (42.10; 47.20) |
4.5-11 |
|
Autumn |
42.80 (41.50;48.20) |
42.80 (39.10; 56.70) |
29.70 (27.10;36.40) |
||
|
Prothrombin activity |
Spring |
90.40 (104.50; 124.20) |
97.10 (95.80;107.70) |
98.60 (91.80;103.00) |
2.5-6 |
|
Autumn |
107.10 (80.60; 122.80) |
132.80 (72.70; 148.30) |
85.80 (71.30;102.30) |
||
|
Prothrombin time |
Spring |
11.70 (11.30;13.50) |
12.50 (11.80;12.60) |
12.40 (12.10;12.90) |
11-12.5 |
|
Autumn |
12.40 (11.30;13.50) |
11.30 (11.10;14.00) |
13.65 (12.80;14.65) |
A—admittance day (Brixia A), H—the day of highest lung involvement of COVID-19 in the ICU (Brixia H), E—the end day of hospitalization in the ICU (Brixia E), APTT—activated partial thromboplastin time. Normal values were replaced by “-“.
3.5. Biochemical blood values evolution on three specific days: Brixia A, H and E.
There was no significant difference between biochemical blood values upon admittance in the two groups, except for calcium values, which were higher in the spring group (spring) (median 9.20 versus 8.80; P=0.02). The CRP values were significantly higher in the autumn group (median 15.00 versus 76.90; P=0.04) on Brixia H day, while the rest of the values were similar in both groups. Only ALT was significantly higher on the Brixia E day in autumn group compared with spring group (median 37.00 versus 76.50; P=0.02) (Table 8).
Table 8. Abnormal and normal biochemical blood values in the two groups on three specific days in the ICU—Brixia A, H and E: median values and interquartile range.
|
Parameters |
Group |
A day |
H day |
E day |
Normal range |
|
ALT |
Spring |
68.00 (21.00; 110.00) |
52.00 (24.00; 72.50) |
37.00 (30.00; 53.00) |
19-25UI/L |
|
Autumn |
30.00 (21.00; 94.00) |
56.00 (28.00; 88.00) |
76.50 (47.00; 115.00) |
||
|
AST |
Spring |
79.50 (31.00; 96.00) |
73.00 (39.50; 85.50) |
37.50 (32.00; 45.00) |
5-40UI/L |
|
Autumn |
44.00 (24.00; 60.00) |
94.00 (44.00; 109.00) |
48.50 (24.00; 86.00) |
||
|
LDH |
Spring |
800.00 (513.00; 816.00) |
722.00 (620.00; 943.00) |
870.50 (665.00; 1420.00) |
140-280U/L |
|
Autumn |
334.00 (334.00; 334.00) |
715.00 (535.00; 895.00) |
511.00 (454.00; 778.00) |
||
|
CRP |
Spring |
12.35 (6.38; 38.30) |
15.00 (6.20; 18.00) |
32.78 (2.15; 38.57) |
<6 |
|
Autumn |
88.10 (87.90; 101.90) |
76.90 (15.80; 122.30) |
22.30 (10.60; 42.30) |
||
|
GGT |
Spring |
47.85 (30.70; 68.85) |
45.60 (24.10; 60.80) |
39.20 (39.10; 80.20) |
0-30U/L |
|
Autumn |
30.30 (18.20; 58.30) |
134.60 (134.60; 134.60) |
214.05 (135.10; 293.00) |
||
|
Albumin |
Spring |
- |
17.15 (2.91; 31.38) |
28.27 (23.75; 32.79) |
3.5-5g/dL |
|
Autumn |
- |
2.44 (2.44; 2.44) |
3.31 (3.26;3.47) |
||
|
Total proteins |
Spring |
- |
5.44 (5.14; 5.63) |
5.31 (4.92; 6.85) |
6-8.3g/dL |
|
Autumn |
- |
5.67 (5.61; 5.73) |
5.91 (5.51; 6.00) |
||
|
Iron |
Spring |
46.80 (34.90; 210.60) |
100.80 (64.30;102.90) |
102.90 (64.30;110.70) |
37-145μg/dL |
|
Autumn |
29.20 (11.00; 47.40) |
32.95 (11.00; 54.90) |
164.90 (164.90; 164.90) |
||
|
Alkaline Reserve |
Spring |
- |
- |
21.60 (17.65; 28.35) |
22-29mmol/L |
|
Autumn |
- |
- |
19.50 (19.50; 19.50) |
A—admittance day (Brixia A), H—the day of highest lung involvement of COVID-19 in the ICU (Brixia H), E—the end day of hospitalization in the ICU (Brixia E), NEUT—neutrophils, APTT—activated partial thromboplastin time, ALT—alanine aminotransferase, AST—aspartate aminotransferase, LDH—lactate dehydrogenase, CRP—C-reactive protein, GGT—gamma glutamyl traspeptidase. Normal values are replaced by “-“.
There was a huge increase in liver enzymes (ALT, AST GGT, LDH) and albumin, reaching higher peaks in the spring group, and remaining high until the end of hospitalization (releasing home/transferring) (Figure 4).
Figure 4. Evolution of some liver enzymes in the two groups on the three Brixia days: A, H and E. All values are above the normal limits.
A—admittance day (Brixia A), H—the day of highest lung involvement of COVID-19 in the ICU (Brixia H), E—the end day of hospitalization in the ICU (Brixia E), ALT—alanine aminotransferase, AST—aspartate aminotransferase, GGT—gamma glutamyl traspeptidase, 1—spring group, 2—autumn group.
In the autumn group there was an important increase in GGT value (median 214.05 versus 39.20 in the spring group) at the end of hospitalization. On the contrary, LDH, which was abnormally high in both groups, was much higher in the spring group than in the autumn group during the entire hospitalization and remained high even at the end. Albumin was abnormally high in the spring group on the H and E days, while it was abnormally low on the H day in the autumn group and then returned to normal. The C-reactive protein was also much higher in the autumn patients compared with spring patients (median 76.90 versus 15.00). These values support our hypothesis that the autumn infection generated more aggressive alterations of blood biochemical values than the spring SARS-CoV-2 infection.
-In Discussion:
In these severe cases of pregnant COVID-19 patients, the inflammatory response (C-reactive protein) was several times higher in autumn patients (when delta variant dominated) upon admission and on the Brixia H day, which probably led to the dramatic decrease in neutrophils during the end days of hospitalization in autumn patients. Meanwhile, the spring infection (when alpha variant dominated) only triggered an inflammatory response, which increased at a constant level throughout the hospitalization. This is in accordance with Fisman [26] and Rangchaikul [27], who described an increased risk of hospitalization, ICU admission and death due to the delta variant compared with the alpha variant in the general population and in the pregnant population. Further, Eid [28] and Zayet [29] found the delta variant more aggressive and dangerous than the previous variants of concern (alpha included).
………
We reported a median Brixia score of 11.3 upon admittance to the ICU in the spring patients, and 8 in the autumn patients, showing that admittance to the ICU was very late in the spring patients, who stayed home as long as they could. Conversely, in autumn, COVID-19 patients went to hospital earlier due to the more intense delta symptoms and perhaps due to the abundant media information advising citizens to consult a physician as soon as they felt symptoms and tested positive for the SARS-CoV-2 antigen. Borghesi [40] also correlated the Brixia H score with the outcomes in COVID-19 patients, showing that patients with an H score under 8 had a good prognosis. We cannot confirm that because our median admittance (A) Brixia score was over 11.3 in spring and over 8 in autumn patients; from there, it went higher to the Brixia H (highest) score. Balbi [41], d’Souza [42] and Au-Yong [43] correlated the Brixia A score with outcomes in COVID-19 patients. Setiawati [44] found that severe cases of COVID-19 pneumonia had a Brixia score over 6, Gatti [45] over 7 upon admission and Boari [46] over 8, but the first and last authors did not specify which Brixia score they used: A, H or E. Similarly, Hoang [47] and Gurtoo [48] also correlated Brixia score with the outcomes in COVID-19 patients but did not specify which Brixia score they used in calculations—either A, H or E score—or if there was more than one chest X-ray taken of each patient. In another study, Boari [49] even correlated the Brixia score with the lung involvement at the posthospitalization follow-up. However, none of them correlated Brixia score with a specific variant of concern, alpha or delta, nor did they describe pregnant patients as we did.
We used the qSOFA score to assess the clinical state of patients on every specific Brixia day—A, H and E—and compared the evolution. In spring patients, the qSOFA score was high upon admittance (A day), decreased slowly until the H day, and then decreased significantly toward the end of hospitalization. In autumn patients, the qSOFA score continued to increase from A day to H day, then decreased dramatically to the end (E day). Moreover, the qSOFA score on admittance (A) day was higher in the spring infection compared with the autumn infection, meaning that the patients’ clinical state was worse in spring patients than in autumn patients upon admittance. During hospitalization, clinical status improved quickly in spring patients (qSOFA score decreased) but continued to deteriorate in autumn patients (qSOFA score increased) towards the H day.
As described above, there was no correlation between the Brixia score and the qSOFA (P=1) score on each particular day—A, H or E—in either the spring nor the autumn group; the Brixia score (radiological evaluation of lung involvement) increased or decreased days later compared to the more reliable qSOFA score, which exactly described the clinical status of the patient that day. However, neither Brixia score nor qSOFA score in one particular day could predict the radiological or clinical evolution of the patient during the next days, in either spring or autumn patients. This is in accordance with Cagino [50], Heldt [51] and Alencar [52], who found no correlation between qSOFA score and outcomes in pregnant or nonpregnant COVID-19 patients. Still, Aashik [53] found the qSOFA score to predict the mortality of COVID-19 patients, while Vikas [54] found the Brixia A score and qSOFA score to predict outcomes in severe forms of pregnant COVID-19 patients.
There was no correlation between either Brixia score or qSOFA score on any particular day (A, H or E) and the blood coagulation factors or biochemical blood values (P=1), in any group. There were no major alterations of the blood cell indices in the two groups, except for the dramatic difference of neutrophils, which was 6.85 in the spring group versus 2.34 (P=0.007) in the autumn group at the end of hospitalization, suggesting that the autumn infection determined a depletion of neutrophils. In the spring group, APTT was slightly over the limit only on the end day, at the same level with the admission day in the autumn group; APTT continued to increase up to the H day in this second group and then decreased, showing that hypercoagulability induced by the autumn infection required more heparin and/or patients were already on heparin when transferred to the ICU.
Since the viral loading of SARS-CoV-2 variants was the most important in the lungs, and second in the liver [55], blood liver enzymes level increased. Jang [56] reported that AST elevated to 126 IU/L and LDH elevated to 409 IU/mL in delta variant pneumonia patients; we observed lower AST and higher LDH in our autumn pregnant patients, when delta variant dominated.
- In the article you refer to” your hypothesis”, but after the modifications of the aim of the study there is no the sentence with this hypothesis. Please read the article carefully so that it is logically consistent with the corrections made.
-We changed and highlighted in YELLOW the hypothesis:
-1. Introduction
Of the recent variants of concern of the SARS-CoV-2 virus, two of them were devastating all around the world: the alpha variant, with a much higher transmissibility than previous ones [1], and the delta variant, which affected even fully vaccinated persons [2]. The delta variant generated more negative outcomes than the alpha in the general population and in the pregnant population. We hypothesize that lung involvement, clinical condition deterioration and blood alterations are also more severe in autumn infection, when the delta variant dominated, compared with spring infections, when the alpha variant dominated, in SARS-CoV-2 critical-infected pregnant patients.
-Then we wrote in the end of Introduction, after the aim of the study:
The aim of the study was to compare the results of chest X-ray examinations, laboratory tests and the results of the qSOFA scale in the group of patients suffering from COVID-19 infection in the spring and the autumn of 2021. We studied the severity of the course of Covid infection in the Brixia scale compared to qSOFA and biochemical results, in order to check our hypothesis that lung involvement, clinical condition deterioration and blood alterations are more severe in autumn infection compared to spring infection, in SARS-CoV-2 critical-infected pregnant patients.
-Then we wrote in Results, page 5:
This was in accordance with the second part of our hypothesis—that is, in autumn group patients, deterioration of the clinical condition was worse than in the spring group patients: their qSOFA score increased despite ICU admission, until H day, and then decreased slowly; meanwhile, in the spring group patients, admittance to the ICU favored slow decrease of qSOFA score.
-Then we wrote in Results, page 7:
There was no significant difference between the complete blood count values of the two groups upon admittance day (Brixia A), except for the red blood cell count, which was significantly lower in the autumn group (P=0.01) but still within normal limits. There were differences between the two groups on the H day: neutrophils, red blood cell count, hemoglobin and mean corpuscular hemoglobin concentration were lower (P=0.01), while mean corpuscular volume was higher (P=0.02) in the autumn group. There were differences between the two groups on the Brixia E day regarding MID leukocytes (P=0.02), mean corpuscular volume (P=0.02), mean platelet volume (P=0.02) and platelet distribution width (P=0.02). The striking difference was the dramatic decrease in neutrophils, from 6.85 in the spring group to 2.34 (P=0.007) in the autumn group at the end of hospitalization, suggesting that the autumn infection determined a depletion of neutrophils; this is in accordance with our initial hypothesis.
-Then, in Results, page 8:
There was a significant difference only on the Brixia E day of hospitalization, when the APTT value was much higher in the spring group (46.80 versus 28.70; P=0.02), showing that the amount of heparin used to the end of hospitalization was much lower in the autumn group versus the spring group, due to the experience gained throughout the spring period. The APTT value in the spring group was slightly over the normal limit only to the end of hospitalization; later, in the autumn group, APTT was higher than normal only on the worst days (H day) and was slightly higher upon admission. Meanwhile, prothrombin time/activity was slightly higher than normal in both groups upon admission and stayed higher on the H day only in the autumn group (Table 7). This was partly in accordance with our hypothesis, since APTT value depended on the amount of heparin used, and the prothrombin time/activity were not dramatically increased.
-Then, in Results, pages 9-10:
In the autumn group there was an important increase in GGT value (median 214.05 versus 39.20 in the spring group) at the end of hospitalization. On the contrary, LDH, which was abnormally high in both groups, was much higher in the spring group than in the autumn group during the entire hospitalization and remained high even at the end. Albumin was abnormally high in the spring group on the H and E days, while it was abnormally low on the H day in the autumn group and then returned to normal. The C-reactive protein was also much higher in the autumn patients compared with spring patients (median 76.90 versus 15.00). These values support our hypothesis that the autumn infection generated more aggressive alterations of blood biochemical values than the spring SARS-CoV-2 infection.
-Then, to the end of Discussion:
In this study, we demonstrated our hypothesis that severe cases of COVID-19 patients behaved worse in autumn 2021, when delta variant dominated, compared with the spring patients, when alpha variant dominated, with regard to lung involvement, coagulation factors and blood enzymes.
- I propose to quote:
Krawczyk P, Jastrzebska A, Szczeklik W, Andres J. Perioperative lung ultrasound pattern changes in patients undergoing gynecological procedures - a prospective observational study. Ginekol Pol. 2021;92(4):271-277. doi: 10.5603/GP.a2021.0017.
Krawczyk P, Jaśkiewicz R, Huras H, Kołak M. Obstetric Anesthesia Practice in the Tertiary Care Center: A 7-Year Retrospective Study and the Impact of the COVID-19 Pandemic on Obstetric Anesthesia Practice. J Clin Med. 2022 Jun 2;11(11):3183.
-We did, as highlighted in References, in YELLOW:
- Krawczyk, P.; Jaśkiewicz, R.; Huras, H.; Kołak, M. Obstetric anesthesia practice in the tertiary care center: a 7-year retrospective study and the impact of the COVID-19 pandemic on obstetric anesthesia practice. J Clin Med. 2022 Jun 2;11(11):3183. doi: 10.3390/jcm11113183.
24.Jang, J.G.; Hur, J.; Hong, K.S.; Lee, W.; Ahn, K.H. Prognostic accuracy of the SIRS, qSOFA, and NEWS for early detection of clinical deterioration in SARS-CoV-2 infected patients. J Korean Med Sci. 2020 Jun 29;35(25):e234. doi: 10.3346/jkms.2020.35.e234.
25.Liu, F.Y.; Sun, X.L.; Zhang, Y. Ge, L.; Wang, J.; Liang, X.; Li, J.F.; Wang, C.L.; Xing, Z.T.; Chhetri, J.; Sun, P.; Chan, P. Evaluation of the risk prediction tools for patients with coronavirus disease 2019 in Wuhan, China: a single-centered, retrospective, observational study. Crit Care Med. 2020 Nov;48(11):e1004-e1011. doi: 10.1097/CCM.0000000000004549.
26.Krawczyk, P.; Jastrzebska, A.; Szczeklik, W.; Andres, J. Perioperative lung ultrasound pattern changes in patients undergoing gynecological procedures- a prospective observational study. Ginekol Pol. 2021;92(4):271-277. doi: 10.5603/GP.a2021.0017. Epub 2021 Mar 23.
Thank you very much!
Submission Date
18 June 2022
Date of this review
20 Jul 2022 13:53:56

This manuscript is a resubmission of an earlier submission. The following is a list of the peer review reports and author responses from that submission.